# PlaySlot: Learning Inverse Latent Dynamics for Controllable Object-Centric Video Prediction and Planning

Angel Villar-Corrales [1]  Sven Behnke [1]

## Abstract

Predicting future scene representations is a crucial task for enabling robots to understand and interact with the environment. However, most existing methods rely on videos and simulations with precise action annotations, limiting their ability to leverage the large amount of available unlabeled video data. To address this challenge, we propose *PlaySlot*, an object-centric video prediction model that infers object representations and latent actions from unlabeled video sequences. It then uses these representations to forecast future object states and video frames. PlaySlot allows the generation of multiple possible futures conditioned on latent actions, which can be inferred from video dynamics, provided by a user, or generated by a learned action policy, thus enabling versatile and interpretable world modeling. Our results show that PlaySlot outperforms both stochastic and object-centric baselines for video prediction across different environments. Furthermore, we show that our inferred latent actions can be used to learn robot behaviors sample-efficiently from unlabeled video demonstrations. Videos and code are available on our project website.

## 1. Introduction

Accurate and flexible world models are crucial for autonomous systems to reason about their surroundings, predict possible future outcomes, and plan their actions effectively. Such models require a structured representation of the world that supports generalization, robustness, and controllability, especially in complex and dynamic scenarios.

[1] Autonomous Intelligent Systems, Computer Science Institute VI – Intelligent Systems and Robotics, Center for Robotics and the Lamarr Institute for Machine Learning and Artificial Intelligence, University of Bonn, Germany . Correspondence to: Angel Villar-Corrales <villar@ais.uni-bonn.de>.

*Proceedings of the $42^{st}$ International Conference on Machine Learning*, Vancouver, Canada. PMLR 267, 2025. Copyright 2025 by the author(s).

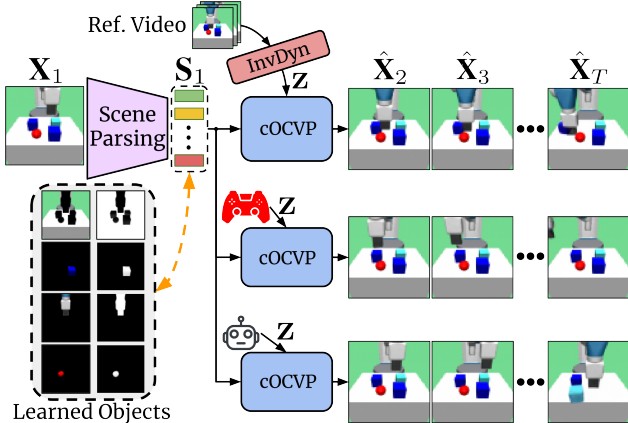

Figure 1: PlaySlot parses an image $\mathbf{X}_1$ into its object components $\mathbf{S}_1$. It then predicts multiple future object states and frames with an object-centric video prediction module (cOCVP) conditioned on latent actions $\mathbf{Z}$, which can be inferred from a reference video with our InvDyn module, provided as input, or generated by a learned action policy.

Humans naturally achieve such understanding by parsing their environment into a *background* and multiple separate *objects*, which can interact with each other and can be recombined to form more complex entities (Johnson, 2018; Kahneman et al., 1992). Neural networks equipped with such compositional inductive biases have the ability to learn structured object-centric representations with desirable properties such as robustness (Bengio et al., 2013; Dittadi et al., 2022), generalization to novel compositions (Greff et al., 2020), transferability to novel tasks (Zhang et al., 2022), and sample efficiency (Mosbach et al., 2025), among others.

Building on these foundations, the field of object-centric learning has made great advances in recent years, progressing from learning object representations in simple synthetic images (Locatello et al., 2020; Burgess et al., 2019) and videos (Kipf et al., 2022; Elsayed et al., 2022), towards more complex real-world scenes (Seitzer et al., 2023; Zadaianchuk et al., 2024). Recently, the field of object-centric video prediction combines these learned object representations with forward-dynamics models and has shown great promise for multiple downstream applications such

as modeling object dynamics (Villar-Corrales et al., 2023; Wu et al., 2023a) or action planning (Yoon et al., 2023; Mosbach et al., 2025). However, such models are currently limited to deterministic environments or rely on videos and simulations with precise action labels to forecast scene dynamics, limiting their ability to leverage unlabeled video data and serve as world models for robotic applications.

In this work, we propose *PlaySlot*, a novel method for controllable video prediction using object-centric representations. PlaySlot learns in a self-supervised manner from video to infer object representations, called slots, and latent action embeddings, which are computed using our proposed *InvDyn* module to capture the scene dynamics. PlaySlot then predicts future video frames conditioned on the inferred object slots and latent actions. At inference time, as illustrated on Fig. 1, PlaySlot parses the observed environment into a set of object slots, each of them representing a different object in the image. Then, PlaySlot forecasts future object states and frames conditioned on past object slots and latent actions, which can be inferred from a video sequence using our proposed InvDyn module, provided by a human, or generated by a learned action policy.

In our experiments, we demonstrate that PlaySlot learns a rich and semantically meaningful action space, enabling accurate video prediction while providing high levels of controllability and interpretability. We show how PlaySlot effectively captures precise robot actions and seamlessly scales to scenes with multiple moving objects or to real-world robotics data, outperforming several controllable video prediction baselines. Moreover, we show that the latent actions inferred by PlaySlot enable sample-efficient learning of robot behaviors from unlabeled demonstrations.

In summary, our contributions are as follows:

- We propose *PlaySlot*—an object-centric video prediction model that infers object representations and latent actions from unlabeled videos, and uses them to forecast future object states and video frames.

- PlaySlot outperforms several video prediction models across diverse robotic environments, while showing superior interpretability and control capabilities.

- The object representations and latent actions inferred by PlaySlot can be used to learn robot behaviors from unlabeled video demonstrations sample efficiently.

## 2. Related Work

**Unsupervised Slot-Based Object-Centric Learning**
Slot-based object-centric representation methods aim to parse in an unsupervised manner an image or video into a set of $N_S$ latent vectors called slots, where each of them binds to a different object in the scene (Greff et al., 2020;

Locatello et al., 2019). Early approaches aimed to learn object representations from synthetic images (Locatello et al., 2020; Singh et al., 2021; Biza et al., 2023) or videos (Kipf et al., 2022; Creswell et al., 2021; Singh et al., 2022) by minimizing a reconstruction objective. To learn meaningful representations from real data, recent slot-based methods leverage weak supervision (Elsayed et al., 2022; Bao et al., 2023), large pretrained transformers (Seitzer et al., 2023; Aydemir et al., 2023; Zadaianchuk et al., 2024), or diffusion models (Jiang et al., 2023; Wu et al., 2023b) These object-centric representations benefit multiple downstream tasks such as reinforcement learning for robotic manipulation (Mosbach et al., 2025; Ferraro et al., 2023) or visual-question-answering (Mamaghan et al., 2025).

**Object-Centric Video Prediction** Object-centric video prediction aims to model the object dynamics and interactions in a video sequence with the goal of forecasting future object states and video frames. Several methods address this task using different architectural priors, including RNNs (Zoran et al., 2021; Assouel et al., 2022), transformers (Villar-Corrales et al., 2023; Wu et al., 2023a; Daniel & Tamar, 2024; Meo et al., 2024) or state-space models (Jiang et al., 2024), attaining a remarkable prediction accuracy on synthetic datasets. Recently, some methods improve the controllability of object-centric video prediction models by conditioning the prediction process on actions (Mosbach et al., 2025) or language captions (Wang et al., 2024; Villar-Corrales et al., 2025; Jeong et al., 2025). However, forecasting future object states without supervision in complex environments still remains an open challenge.

**Learning Latent Actions from Unlabeled Videos**
Videos provide abundant information about dynamics and activities, but often lack the action labels necessary for learning behaviors from video. To address this challenge, some methods train a latent policy directly from observations by learning a discrete latent action space and sampling the actions that minimize a reconstruction error (Edwards et al., 2019; Struckmeier & Kyrki, 2023). Another group of methods, to which PlaySlot belongs, learns inverse dynamics from unlabeled videos by predicting latent actions given pairs of observations, and uses them for learning behaviors for video games and robot simulations (Ye et al., 2022; Brandfonbrener et al., 2024; Schmidt & Jiang, 2024), as pretraining for Vision-Language-Action models (Ye et al., 2025) or for learning robot policies (Cui et al., 2024; Tian et al., 2025).

Latent action models have also been used for conditional video prediction. The most similar method to ours is *CADDY* (Menapace et al., 2021; 2022), which learns latent actions from a collection of unlabeled videos from

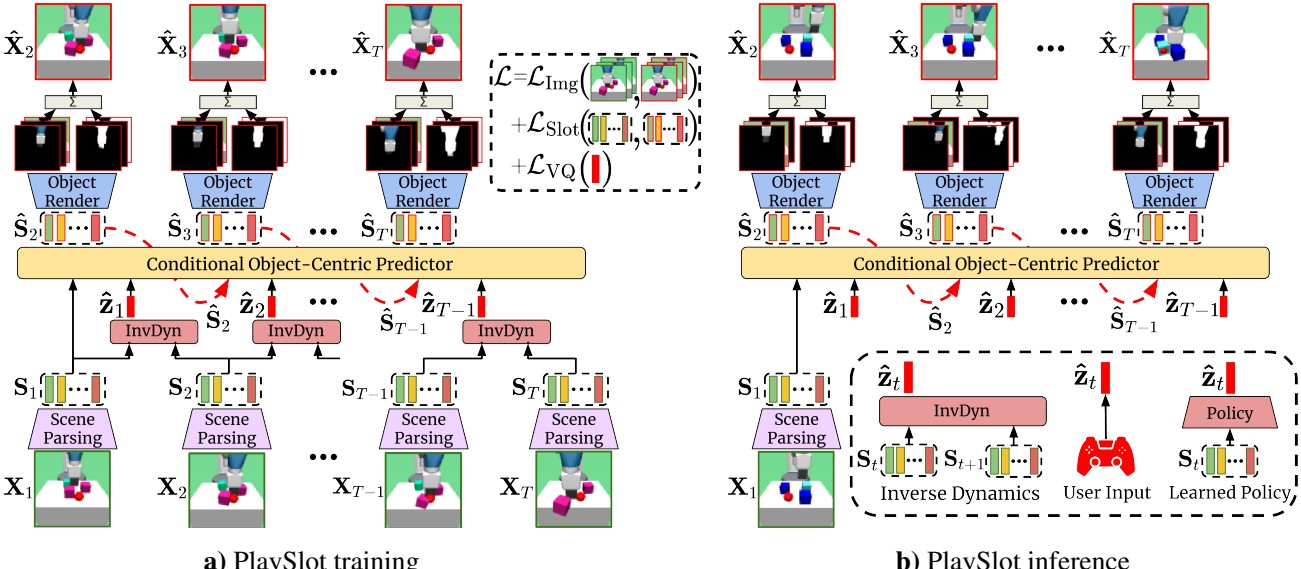

**a)** PlaySlot training      **b)** PlaySlot inference

Figure 2: Overview of PlaySlot training and inference processes. **(a)** PlaySlot is trained given unlabeled video sequences by inferring object representations $\mathbf{S}_t$ and latent actions $\hat{\mathbf{z}}_t$, and using these representations to autoregressively forecast future video frames and object states. **(b)** PlaySlot autoregressively forecasts future frames conditioned on a single frame $\mathbf{X}_1$ and latent actions $\hat{\mathbf{z}}$, which can be inferred from observations, provided by a user, or output by a learned action policy.

a single domain and uses the latent actions as conditioning signal for predicting future frames. At inference time, CADDY maps user inputs to the latent space for playable video generation. Building upon this same principle, *Genie* (Bruce et al., 2024) proposes a foundation world model for playable video generation on diverse environments. However, both CADDY and Genie operate on holistic scene representations, which are limited for tasks that require relational reasoning, often struggle to model object relationships and interactions, and require human supervision to generalize to scenes with multiple moving agents.

## 3. PlaySlot

We propose *PlaySlot*, a novel framework for controllable object-centric video prediction from unlabeled video sequences. Fig. 2a) illustrates the training process in PlaySlot, as well as its main four components. Namely, given $T$ video frames $\mathbf{X}_{1:T}$, our model employs as *Scene Parsing module* (Sec. 3.1) that decomposes these images into object representations, called slots, $\mathbf{S}_{1:T} = (\mathbf{S}_1, ..., \mathbf{S}_T)$, where $\mathbf{S}_t = (\mathbf{s}_t^1, ..., \mathbf{s}_t^{N_\mathbf{S}}) \in \mathbb{R}^{N_\mathbf{S} \times D_\mathbf{S}}$ is the set of $D_\mathbf{S}$-dimensional object slots parsed from frame $\mathbf{X}_t$. For each consecutive pair of frames, PlaySlot employs an *Inverse Dynamics* (InvDyn) module (Sec. 3.2) in order to estimate latent action embeddings $\hat{\mathbf{z}}_t$ that encode the actions taken by the agents in the scene between every consecutive pair of frames. The *Conditional Object-Centric Predictor* (cOCVP) (Sec. 3.3) forecasts future object states conditioned on past slots and latent actions estimated by

InvDyn. Finally, the *object rendering* module decodes the object slots to render object images and masks, which can be combined via a weighted sum to render video frames.

At inference time, as shown in Fig. 2b), PlaySlot autoregressively predicts multiple possible sequence continuations conditioned on the initial object slots and latent action embeddings, which can be estimated by InvDyn, provided by a user, or generated by a learned action policy.

### 3.1. Object-Centric Representation Learning

PlaySlot employs SAVi (Kipf et al., 2022), a recursive encoder-decoder model with a structured bottleneck of $N_\mathbf{S}$ permutation-equivariant object slots, to parse a sequence of video frames $\mathbf{X}_{1:T}$ into their object components $\mathbf{S}_{1:T}, \mathbf{S}_t \in \mathbb{R}^{N_\mathbf{S} \times D_\mathbf{S}}$. The slots $\mathbf{S}_0$ are sampled from a learned distribution and recursively refined to bind to the objects in the video frames. At time step $t$, SAVi encodes the corresponding input $\mathbf{X}_t$ into feature maps $\mathbf{h}_t \in \mathbb{R}^{L \times D_h}$, which are fed to Slot Attention (Locatello et al., 2020) to iteratively refine the previous slot representations conditioned on the current features. Slot Attention performs cross-attention between the image features and slots with the attention weights normalized over the slot dimension, encouraging competition between slots so as to represent each feature location:

$$\mathbf{A} = \underset{N_\mathbf{S}}{\text{softmax}} \left( \frac{q(\mathbf{S}_{t-1}) \cdot k(\mathbf{h}_t)^T}{\sqrt{D_\mathbf{S}}} \right) \in \mathbb{R}^{N_\mathbf{S} \times L}, \quad (1)$$

where $q$ and $k$ are linear projections. The slots are then independently updated via a shared Gated Recurrent

Unit (Cho et al., 2014) (GRU) followed by a residual MLP:

$$\mathbf{S}_t = \text{GRU}(\mathbf{A} \cdot v(\mathbf{h}_t), \mathbf{S}_{t-1}) , \quad \mathbf{A}_{n,l} = \frac{\mathbf{A}_{n,l}}{\sum_{i=0}^{L-1} \mathbf{A}_{n,i}}, \quad (2)$$

where $v$ is a linear projection. The steps described in Equations (1) and (2) can be repeated multiple times with shared weights to iteratively refine the slots and obtain an accurate object-centric representation of the scene.

To map the object representations back to images, SAVi independently decodes each slot in $\mathbf{S}_t$ with a Spatial Broadcast Decoder (Watters et al., 2019) ($\mathcal{D}_{\text{SAVi}}$) to render an object image and mask, which can be normalized and combined via a weighted sum to render the reconstructed frame:

$$\mathbf{o}_t^n, \mathbf{m}_t^n = \mathcal{D}_{\text{SAVi}}(\mathbf{s}_t^n), \quad (3)$$

$$\hat{\mathbf{X}}_t = \sum_{n=1}^{N_{\mathbf{S}}} \mathbf{o}_t^n \cdot \tilde{\mathbf{m}}_t^n \quad \text{with} \quad \tilde{\mathbf{m}}_t^n = \underset{N_{\mathbf{S}}}{\text{softmax}}(\mathbf{m}_t^n). \quad (4)$$

### 3.2. Learning Inverse Dynamics

In general, future frames depend not only on previous observations, but also on other variables, such as robot actions. We propose an inverse dynamics module (InvDyn) that estimates, given the object slots from two consecutive time steps, latent action embeddings $\hat{\mathbf{z}} \in \mathbb{R}^{D_{\mathbf{z}}}$ that encode the actions taken by the agents between such time steps:

$$\hat{\mathbf{z}}_t = \text{InvDyn}(\mathbf{S}_t, \mathbf{S}_{t+1}). \quad (5)$$

#### 3.2.1. ACTION PARAMETERIZATION

The parameterization of the latent actions determines the complexity of transitions that can be modeled, as well as the degree of control that we have over the predictions. On the one hand, learning a finite set of latent actions enables controllable video prediction while limiting the complexity of the dynamics that such actions can explain. On the other hand, continuous latent vectors offer greater flexibility to model complex transitions between frames, but at the cost of reduced interpretability and control.

As a compromise between these two approaches, inspired by Menapace et al. (2021), we propose a hybrid parameterization of the latent action $\hat{\mathbf{z}}_t$ as the sum of a discrete $\mathbf{p}_t$ and a continuous $\mathbf{v}_t$ component: $\hat{\mathbf{z}}_t = \mathbf{p}_t + \mathbf{v}_t$. The discrete component $\mathbf{p}_t$, denoted as *action prototype*, determines the high-level action taking place (e.g. move left, go up), whereas the continuous *action variability* $\mathbf{v}_t$ captures stochastic and fine-grained variations, allowing the model to interpolate between prototypes and account for environmental uncertainty. This hybrid design enables the model to learn expressive dynamics while maintaining control and interpretability.

#### 3.2.2. INVDYN MODULE

We propose two variants of our inverse dynamics module for inferring latent actions from object-centric representations. InvDyn$_{\text{S}}$ processes object slots $\mathbf{S}_t$ alongside a learnable token `[ACT]` using a transformer encoder $f_{\mathbf{z}}$, producing a single latent action $\hat{\mathbf{z}}_t$ that represents the agent's behavior. This formulation is particularly well-suited for single-agent environments. In contrast, InvDyn$_{\text{M}}$ processes each slot with a shared MLP, producing $N_{\mathbf{S}}$ latent action embeddings $\hat{\mathbf{Z}}_t = \{\hat{\mathbf{z}}_t^1, ..., \hat{\mathbf{z}}_t^{N_{\mathbf{S}}}\}$, where each vector represents the action of a specific object in the scene. Below we explain the process for computing latent actions using InvDyn$_{\text{S}}$, which follows a similar procedure to that of InvDyn$_{\text{M}}$.

Following Menapace et al. (2021), we adopt a probabilistic formulation where InvDyn predicts the posterior distribution of scene dynamics, modeled as Gaussian:

$$\boldsymbol{\mu}_{\mathbf{d}_t}, \boldsymbol{\sigma}_{\mathbf{d}_t}^2 = f_{\mathbf{z}}(\mathbf{S}_t, \texttt{[ACT]}). \quad (6)$$

The distribution of latent actions $\hat{\mathbf{z}}_t$ is then defined as the difference between the distributions of scene dynamics from two consecutive time steps:

$$\hat{\mathbf{z}}_t \sim \mathcal{N}(\boldsymbol{\mu}_{\mathbf{z}_t}, \boldsymbol{\sigma}_{\mathbf{z}_t}^2) \quad \text{with} \quad \begin{cases} \boldsymbol{\mu}_{\mathbf{z}_t} = \boldsymbol{\mu}_{\mathbf{d}_{t+1}} - \boldsymbol{\mu}_{\mathbf{d}_t}, \\ \boldsymbol{\sigma}_{\mathbf{z}_t}^2 = \boldsymbol{\sigma}_{\mathbf{d}_{t+1}}^2 + \boldsymbol{\sigma}_{\mathbf{d}_t}^2, \end{cases} \quad (7)$$

from which latent actions $\hat{\mathbf{z}}_t$ are sampled.

To prevent the model from trivially encoding future states into $\hat{\mathbf{z}}_t$, we regularize the latent action space by enforcing an information bottleneck. Specifically, we constrain the latent action space to be low dimensional, i.e., $D_{\mathbf{z}} << D_{\mathbf{S}}$. Furthermore, as discussed in Sec. 3.2.1, we parameterize the latent actions as the sum of a discrete action prototype $\mathbf{p}_t$ and a continuous action variability embedding $\mathbf{v}_t$, where $\mathbf{p}_t$ is obtained by vector-quantizing the latent actions $\hat{\mathbf{z}}_t$ ($\mathbf{p}_t = \text{VQ}(\hat{\mathbf{z}}_t)$). We empirically verify that the information bottleneck enforced by vector quantization achieves comparable performance to the one proposed by (Menapace et al., 2021), while requiring significantly fewer hyperparameters.

Our hybrid latent action parameterization ensures that our InvDyn module encodes only the essential dynamics, effectively capturing the agent's interaction with the scene while learning semantically meaningful action prototypes. Moreover, this hybrid factorization improves the controllability and interpretability of the prediction process while maintaining the ability to model complex scene dynamics.

### 3.3. Conditional Object-Centric Prediction

We employ a transformer-based (Vaswani et al., 2017) module to autoregressively predict future object slots conditioned on past object states and latent actions.

Our proposed predictor, cOCVP, is a transformer encoder with $N_{\text{Pred}}$ layers. At each time step $t$, cOCVP takes as input all previous slots $\mathbf{S}_{1:t}$, action prototypes $\mathbf{p}_{1:t}$ and variability embeddings $\mathbf{v}_{1:t}$, all of which are first linearly projected into a shared token dimensionality. The slots are then conditioned by adding them with the corresponding projected action prototype and variability embeddings. Additionally, we incorporate sinusoidal positional encodings such that all slots from the same time step receive the same encoding, thus preserving the inherent permutation equivariance of the objects.

cOCVP forecasts the future slots $\hat{\mathbf{S}}_{t+1}$ by jointly modeling the object dynamics and interactions from the past object slots conditioned on the inferred latent actions. This process is summarized as:

$$\hat{\mathbf{S}}_{t+1} = \text{cOCVP}(f_{\text{S}}(\mathbf{S}_{1:t}) + f_{\text{p}}(\mathbf{p}_{1:t}) + f_{\text{v}}(\mathbf{v}_{1:t})), \quad (8)$$

where $f_{\text{S}}$, $f_{\text{p}}$ and $f_{\text{v}}$ are learned linear layers.

The prediction process can be initiated from the slots of a single reference frame $\mathbf{S}_1$ and the corresponding inferred latent actions $\hat{\mathbf{z}}_1$. This process is repeated autoregressively, with the predicted slots being appended to the input at each subsequent time step, allowing the generation of future object representations for a desired number of time steps $T$.

### 3.4. Learning Behaviors from Unlabeled Videos

Using a pretrained InvDyn module, we aim to learn a latent policy from unlabeled video expert demonstrations—without the need for action or reward information. For this purpose, we first use InvDyn to annotate a dataset of unlabeled expert demonstrations with latent actions that capture the underlying scene dynamics. Subsequently, we train a latent policy model $f_{\pi}$ via behavior cloning, regressing the inferred latent actions from the object slots observed at the corresponding time step.

At inference time, starting from a single observation $\mathbf{X}_1$, PlaySlot computes the corresponding object slots $\mathbf{S}_1$ and uses the learned policy to predict a latent action $\hat{\mathbf{z}}_1$, which is decomposed into an action prototype $\mathbf{p}_1 = \text{VQ}(\hat{\mathbf{z}}_1)$ and a variability embedding $\mathbf{v}_1 = \hat{\mathbf{z}}_1 - \mathbf{p}_1$. These representations are fed to cOCVP to forecast the subsequent set of object slots $\hat{\mathbf{S}}_2$. This process is repeated autoregressively, allowing the learned behavior to unfold within the model's latent imagination.

To enable the execution of latent actions in a simulator, we introduce an action decoder $\mathcal{D}_{\text{a}}$ that maps the predictions of the latent policy $f_{\pi}$ to real-world actions. This module, implemented as a three-layer MLP, is trained on a small set of action-labeled data to translate latent actions inferred by InvDyn into the corresponding executable actions. Importantly, the action decoder is only needed to execute latent

actions in the simulator for evaluation purposes, and the core training process of PlaySlot is fully unsupervised, operating entirely on unlabeled, action-free videos.

Our approach shares similarities with Schmidt & Jiang (2024), who learn policies from video using discrete latent actions in simple game environments. In contrast, PlaySlot leverages a more expressive action representation and a conditional object-centric decoder, enabling the learning of more complex robot behaviors in realistic environments.

### 3.5. Training

We differentiate three different training stages in PlaySlot. We first train SAVi to parse video frames into object-centric representations by minimizing a reconstruction loss:

$$\mathcal{L}_{\text{SAVi}} = \sum_{t=1}^{T} ||\mathcal{D}_{\text{SAVi}}(\mathcal{E}_{\text{SAVi}}(\mathbf{X}_t)) - \mathbf{X}_t||_2^2, \quad (9)$$

where $\mathcal{E}_{\text{SAVi}}$ and $\mathcal{D}_{\text{SAVi}}$ correspond to the scene parsing and object rendering modules, respectively.

Second, given the pretrained SAVi model, we jointly train InvDyn and cOCVP by minimizing a combined loss:

$$\mathcal{L}_{\text{PlaySlot}} = \sum_{t=2}^{T+1} \lambda_{\text{Img}}\mathcal{L}_{\text{Img}} + \lambda_{\text{Slot}}\mathcal{L}_{\text{Slot}} + \lambda_{\text{VQ}}\mathcal{L}_{\text{VQ}}, \quad (10)$$

$$\mathcal{L}_{\text{Img}} = ||\hat{\mathbf{X}}_t - \mathbf{X}_t||_2^2, \quad (11)$$

$$\mathcal{L}_{\text{Slot}} = ||\hat{\mathbf{S}}_t - \mathcal{E}_{\text{SAVi}}(\mathbf{X}_t)||_2^2, \quad (12)$$

$$\mathcal{L}_{\text{VQ}} = ||\text{sg}[\hat{\mathbf{z}}_t] - \mathbf{p}_t|| + 0.25 \cdot ||\hat{\mathbf{z}}_t - \text{sg}[\mathbf{p}_t]||, \quad (13)$$

where $\text{sg}$ is the stop-gradient operator. $\mathcal{L}_{\text{Img}}$ measures the future frame prediction error, $\mathcal{L}_{\text{Slot}}$ aligns the predicted object slots with the actual object-centric representations, and $\mathcal{L}_{\text{VQ}}$ encourages the learning of meaningful action prototypes while regularizing the latent actions to align with their prototypes (Van Den Oord & Vinyals, 2017). We do not employ teacher forcing, enabling the predictor model to learn to handle its own imperfect predictions.

Finally, the policy model $f_{\pi}$ and action decoder $\mathcal{D}_{\text{a}}$ are trained to regress the inferred latent actions $\hat{\mathbf{z}}$ and ground truth actions $\mathbf{a}$, respectively:

$$\mathcal{L}_{f_{\pi}} = \sum_{t=1}^{T} ||f_{\pi}(\mathbf{S}_t) - \hat{\mathbf{z}}_t||, \quad (14)$$

$$\mathcal{L}_{\mathcal{D}_{\text{a}}} = \sum_{t=1}^{T} ||\mathcal{D}_{\text{a}}(\hat{\mathbf{z}}_t) - \mathbf{a}_t||. \quad (15)$$

Table 1: Quantitative evaluation of several object-centric (OC) and controllable (Cont.) video prediction models. Given six seed frames, all models predict the subsequent 15 frames. PlaySlot achieves the best results in datasets that require modeling object interactions (BlockPush) or feature multiple moving objects (GridShapes), while maintaining competitive performance on ButtonPress and Sketchy. Best two results are highlighted in boldface and underlined, respectively.

| Model | OC | Cont. | BlockPush | | | ButtonPress | | | GridShapes$_{2Objs}$ | | | Sketchy | | |
|---|---|---|---|---|---|---|---|---|---|---|---|---|---|---|
| | | | PSNR↑ | SSIM↑ | LPIPS↓ | PSNR↑ | SSIM↑ | LPIPS↓ | PSNR↑ | SSIM↑ | LPIPS↓ | PSNR↑ | SSIM↑ | LPIPS↓ |
| SVG | ✗ | ✓ | 20.96 | 0.898 | 0.096 | **32.23** | **0.950** | **0.011** | 38.10 | 0.988 | 0.002 | 24.24 | 0.813 | 0.076 |
| CADDY | ✗ | ✓ | 21.18 | **0.901** | 0.090 | 24.58 | 0.853 | 0.028 | 39.16 | 0.986 | 0.006 | 23.65 | 0.809 | **0.059** |
| SlotFormer | ✓ | ✗ | 17.22 | 0.729 | 0.134 | 19.52 | 0.762 | 0.111 | 19.74 | 0.795 | 0.149 | 21.17 | 0.751 | 0.091 |
| OCVP | ✓ | ✗ | 17.26 | 0.751 | 0.134 | 19.55 | 0.762 | 0.115 | 18.86 | 0.791 | 0.154 | 21.23 | 0.751 | 0.092 |
| PlaySlot | ✓ | ✓ | **21.41** | 0.890 | **0.066** | 26.03 | 0.878 | 0.025 | **54.09** | **0.996** | **0.001** | **24.43** | **0.815** | 0.063 |

## 4. Experiments

### 4.1. Experimental Setup

#### 4.1.1. DATASETS

We evaluate our method on four environments with distinct characteristics. Further details are provided in Appendix C.

**ButtonPress** This environment, based on *MetaWorld* (Yu et al., 2020), features a Sawyer robot arm that must press a red button. This environment depicts a non-object centric task involving complex shapes and textures.

**BlockPush**: This environment, inspired by Li et al. (2020), features a robot arm and a table with multiple uni-colored cubes. The robot must push the block of distinct color into the location specified by a red target. This task evaluates relational reasoning and modeling of object collisions.

**GridShapes:** This dataset features two simple 2D shapes moving in grid-like patterns on a colored background. The shapes randomly change direction, introducing stochasticity to their motion. This dataset benchmarks a model's ability to jointly predict the motion of multiple moving agents.

**Sketchy:** Sketchy (Cabi et al., 2020) is a real-world robotics dataset in which a robotic gripper interacts with diverse objects of different shape and color. This dataset evaluates the model performance on real-world robotics data.

#### 4.1.2. IMPLEMENTATION DETAILS

Our model is implemented in PyTorch (Paszke et al., 2017) and trained on a single NVIDIA A100 GPU. PlaySlot uses SAVi (Kipf et al., 2022) with 128-dimensional object slots, as well as a convolutional encoder and spatial broadcast decoder as scene parsing and rendering modules, respectively. The conditional predictor and inverse dynamics modules are transformer encoders with four layers and a token dimension of 256. For ButtonPress, BlockPush and Sketchy we use the InvDyn$_S$ variant with eight distinct 16-dimensional action prototypes, whereas for GridShapes we use InvDyn$_M$ with five 8-dimensional prototypes. Further implementation details are provided in Appendix B.

### 4.2. Video Prediction

We compare PlaySlot for video prediction with the object-centric baselines SlotFormer (Wu et al., 2023a) and OCVP-Seq (Villar-Corrales et al., 2023), the stochastic video prediction model SVG-LP (Denton & Fergus, 2018) and the playable video generation model CADDY (Menapace et al., 2021). For fair comparison, all models are trained with six seed frames to predict the subsequent eight, and evaluated for 15 predictions. For CADDY, PlaySlot and SVG, we predict future frames conditioned on latent actions or vectors inferred from the ground truth sequence. Additionally, on BlockPush and ButtonPress datasets all models are trained using sequences with random exploration policies, and evaluated on expert demonstrations.

We evaluate the quality of the predicted frames using PSNR, SSIM (Wang et al., 2004) and LPIPS (Zhang et al., 2018). A quantitative comparison of the methods is presented in Tab. 1. As expected, deterministic object-centric models (i.e. SlotFormer and OCVP) perform poorly, as they cannot infer the agent's actions and simply average over multiple possible futures. Our proposed method outperforms all other models on both the BlockPush and GridShapes datasets, demonstrating PlaySlot's superior ability to forecast future video frames in environments involving multiple object interactions and moving agents, respectively. On Sketchy, which features real robotics videos, we observe that PlaySlot, CADDY and SVG achieve similar performance, with PlaySlot slightly outperforming the baselines. While the margin of improvement is modest, these results suggest that object-centric models like PlaySlot are well-suited for controllable video prediction and can serve as world models in real robotic scenarios.

Fig. 3 depicts a qualitative comparison of the best performing methods on the ButtonPress and BlockPush datasets, respectively. On the ButtonPress dataset, as shown in Fig. 3a), all methods accurately model the motion of the robot arm. However, on the more complex BlockPush task, depicted in Fig. 3b), SVG and CADDY fail to model the object collisions, leading to blurriness and vanishing objects. In contrast, PlaySlot maintains sharp object represen-

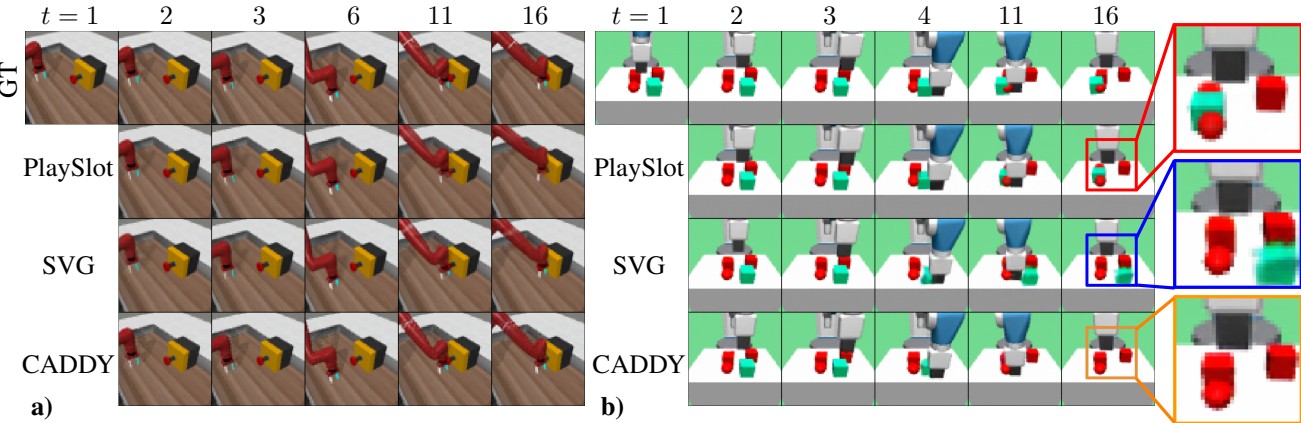

Figure 3: Qualitative comparison on (a) ButtonPress and (b) BlockPush datasets. PlaySlot accurately predicts the scene dynamics, whereas baselines fail to predict object interactions, leading to blurriness and disappearing objects.

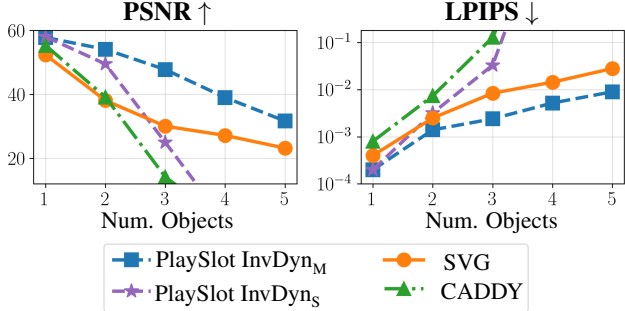

Figure 4: Quantitative results on the GridShapes dataset with different numbers of objects. PlaySlot outperforms the baselines, particularly for a higher number of objects.

tations and correctly models interactions between objects, leading to accurate frame predictions. Further qualitative evaluations are provided in Appendix E.

### 4.3. Model Analysis

**Impact of Number of Moving Objects:** We evaluate the performance of our method for different numbers of moving objects. For this purpose, we train two PlaySlot variants with the $InvDyn_S$ and $InvDyn_M$ inverse dynamics modules, respectively, and compare them with the SVG and CADDY baselines on several variants of the GridShapes dataset featuring a different number of objects, ranging from one to five moving shapes. The results are shown in Fig. 4. CADDY and PlaySlot with $InvDyn_S$, which encode scene dynamics using a single latent action, perform strongly when forecasting one or two objects, but experience a sharp drop in performance as the number of objects increases. SVG scales to multiple moving objects but encodes all dynamics into a single distribution, limiting its flexibility and control. In contrast, PlaySlot with $InvDyn_M$ uses a latent action per object, seamlessly scaling to a large number of moving agents by individually modeling the motion of each object, thus outperforming all baselines.

Table 2: Quantitative comparison of PlaySlot with variants using continuous and discrete latent actions, as well as an oracle model with access to the ground truth actions.

| | | Results | | |
|---|---|---|---|---|
| **RobotDB** | **Model Variant** | PSNR↑ | SSIM↑ | LPIPS↓ |
| **Random Exploration** | PlaySlot | 26.64 | 0.944 | **0.016** |
| | w/ Cont. $\hat{z}$ | 26.24 | 0.924 | 0.019 |
| | w/ Discrete $\hat{z}$ | 20.60 | 0.849 | 0.040 |
| | w/ GT Actions | **27.77** | **0.955** | **0.016** |
| **Expert Demos.** | PlaySlot | 21.41 | 0.890 | 0.065 |
| | w/ Cont. $\hat{z}$ | 22.00 | 0.900 | 0.061 |
| | w/ Discrete $\hat{z}$ | 18.00 | 0.791 | 0.109 |
| | w/ GT Actions | **22.30** | **0.904** | **0.058** |

**Action Representation:** In Tab. 2 we compare the hybrid latent action representation used in PlaySlot with three different variants that use continuous latent actions, a discrete set of latent actions, and an oracle variant with access to ground truth actions. Additionally, we evaluate the models on two different BlockPush variants, including a set with random exploration sequences, similar to the training distribution, and another set featuring expert demonstrations.

On the random exploration set, PlaySlot with a hybrid latent action performs comparably to the oracle, highlighting the ability of our InvDyn module to infer latent dynamics from unlabeled sequences. However, PlaySlot's performance drops on the expert demonstrations, with the variant using unconstrained continuous latent actions outperforming it. We attribute this to the large discrepancy between the action distribution of expert sequences and the training data, which challenges the generalization of the learned action prototypes. In both cases, the variant with only discrete latent actions performs the worst, likely due to its limited flexibility in modeling the fine-grained dynamics of robot motion. Nonetheless, our hybrid action representation remains competitive, achieving performance close to the best methods despite the distribution shift.

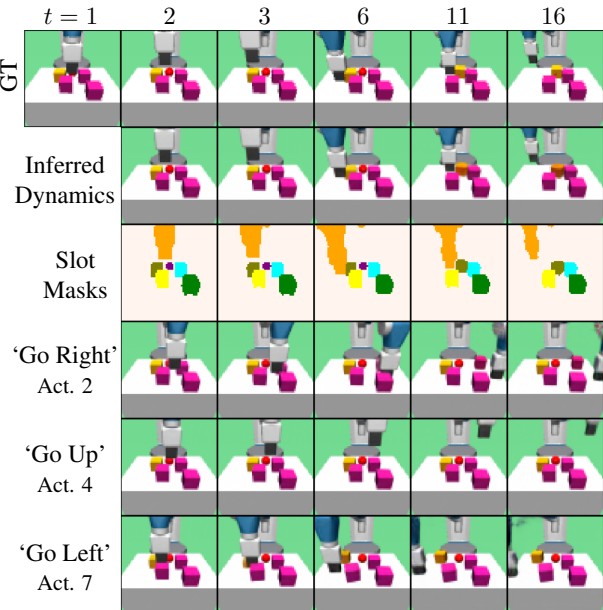

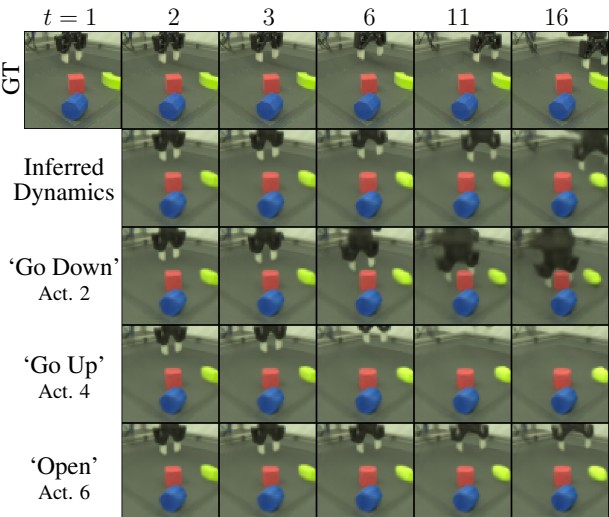

Figure 6: Qualitative results on a real-world robotics sequence. PlaySlot accurately predicts possible futures conditioned on a single reference frame and latent actions.

Figure 5: PlaySlot predictions given different latent actions, including inferred inverse dynamics and three action prototypes. PlaySlot learns accurate object-centric representations and semantically consistent action prototypes.

**Learned Action Prototypes:** Fig. 5 illustrates the effect of different action prototypes learned by PlaySlot on the BlockPush dataset. Given a single seed frame, we forecast 15 frames by repeatedly conditioning the predictor on the same action prototype. We also visualize the predictions obtained using the latent actions inferred by our inverse dynamics module from the ground truth sequence. PlaySlot effectively learns to infer precise robot actions from visual observations, as well as the physics of interacting objects. Additionally, Fig. 5 shows that PlaySlot learns consistent and semantically meaningful actions, such as moving the robot towards the right (action 2), left (action 7), or upwards (action 4). Finally, the instance segmentation maps, obtained by assigning a different color to each slot mask, demonstrate how PlaySlot distinctly represents each object into a separate slot, clearly disentangling each individual block, the robot and the background.

**Real-World Robotic Videos:** We validate the applicability of PlaySlot to real-world robotic videos using the Sketchy (Cabi et al., 2020) dataset. Fig. 6 presents a qualitative assessment demonstrating PlaySlot's ability to accurately infer the scene's inverse dynamics from a real robotic demonstration. Starting from a single reference frame, our model reconstructs the ground truth sequence by predicting the robot's movement, rotation, and gripper opening. Furthermore, PlaySlot learns semantically meaningful and temporally consistent action prototypes that capture diverse robot behaviors such as opening the gripper (action 6), moving downwards (action 2), or upwards (action 4).

### 4.4. Learning Behaviors from Expert Demonstrations

We evaluate the quality of PlaySlot's object-centric representations and inferred latent actions for a downstream behavior learning task. To this end, we train a policy model and an action decoder, as described in Sec. 3.4, to imitate ButtonPress and BlockPush behaviors from a limited set of expert demonstrations. Furthermore, we train the models using three different random seeds and show the mean success rate and its standard deviation.

**ButtonPress Behavior** We compare our learned policy, which imitates the ButtonPress behavior, against oracle baselines with access to all expert demonstrations. These models directly regress the ground-truth actions from object-centric slots (Slot Oracle), ResNet feature (ResNet Oracle), and from feature maps output by the CADDY encoder (CADDY Oracle), respectively. In contrast, PlaySlot autoregressively predicts latent actions within its latent imagination before decoding them into executable actions. We also evaluate a modified CADDY variant with a lightweight policy and action decoder that maps observed features to real-world actions.

As shown in Fig. 7a), PlaySlot consistently outperforms CADDY across all training regimes. With as few as 200 demonstrations, PlaySlot matches the performance of the CADDY Oracle, which has direct access to more expert demonstrations and ground-truth actions. As the number of demonstrations increases, PlaySlot continues to improve, approaching the performance of the oracle models, while CADDY struggles to generalize. These findings highlight the effectiveness of PlaySlot's object-centric representations in learning robot behaviors from limited expert data. Additionally, we observe that slot-based models outper-

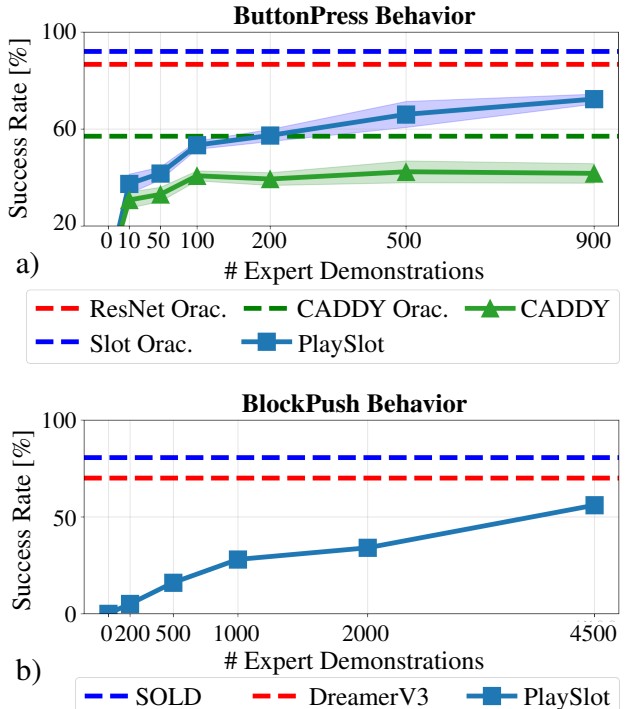

Figure 7: Success rate (%) as a function of available expert demonstrations on the a) ButtonPress and b) BlockPush environments. Object-centric models (Slot Oracle, PlaySlot and SOLD) outperform their counterparts. PlaySlot consistently improves with more demonstrations.

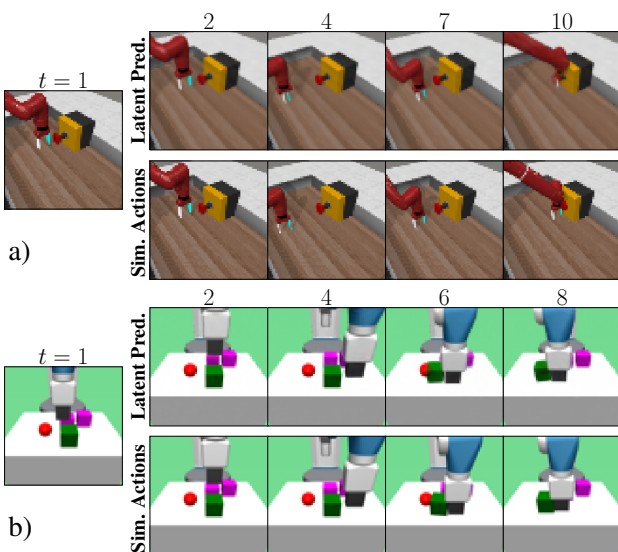

Figure 8: Predicted frames using latent actions generated by the learned latent policy, and sequences simulated by executing the decoded latent actions for a) ButtonPress and b) BlockPush learned behaviors.

form their holistic counterparts, further emphasizing the advantage of structured object-centric representations for learning robot behaviors.

**BlockPush Behavior** We evaluate a policy learned using PlaySlot on the challenging BlockPush task, which requires reasoning over object properties. Our learned policy imitates the behavior from a limited number of expert demonstrations, with only approximately 80% being successful. The imperfect nature of the expert data adds an extra layer of difficulty to the learning process.

We compare PlaySlot to two model-based reinforcement learning baselines with holistic (DreamerV3 (Hafner et al., 2023)) and object-centric (SOLD (Mosbach et al., 2025)) latent spaces, both of which learn the robot behavior by interacting with the environment. As shown in Fig. 7b), PlaySlot closes the gap with the baseline models as the number of available demonstrations increases, demonstrating its ability to generalize from limited data.

**Visualization of Learned Behaviors** Fig. 8 illustrates the learned latent policies in the (a) ButtonPress and (b) BlockPush environments, respectively. The top row in each sequence (labeled *Latent Pred.*) shows predicted trajectories within PlaySlot's latent imagination, where the model,

starting from a single reference frame, autoregressively generates latent actions using the policy model and predicts future scene states in the latent space. The bottom row (labeled *Sim. Actions*) visualizes the simulated execution of the decoded latent actions in the respective environments. PlaySlot learns to solve both tasks within its latent imagination, successfully reasoning over object properties and producing coherent sequences of latent actions that can be decoded into executable motions.

## 5. Conclusion

We introduced PlaySlot, a novel framework for controllable object-centric video prediction. PlaySlot parses video frames into object slots, infers the scene's inverse dynamics, and predicts future object states and video frames by modeling the object dynamics and interactions, conditioned on inferred latent actions. Through extensive experiments, we demonstrated that PlaySlot learns a semantically rich and meaningful action space, allowing for accurate video frame predictions. Our method outperforms several baseline models, offering superior controllability and interpretability. We further demonstrated that the learned object-centric representations and inferred latent actions can be utilized to predict future frames with precise robot control, while also enabling learning complex robot behaviors from limited, unlabeled demonstrations—outperforming methods that rely on holistic scene encoding. These results highlight PlaySlot as a versatile and interpretable world model, well-suited for a wide range of tasks in autonomous systems.

## Acknowledgment

This work was funded by grant BE 2556/16-2 (Research Unit FOR 2535 Anticipating Human Behavior) of the German Research Foundation (DFG). Computational resources were provided by the German AI Service Center WestAI.

## Impact Statement

This paper presents work whose goal is to advance the field of machine learning by introducing PlaySlot—an object-centric video prediction model that forecasts future video frames conditioned on inferred object-centric representations and latent actions, enhancing its interpretability and controllability, as well as learning representations that can be used for action planning. This advancement is particularly relevant for domains such as robotics and autonomous systems, where understanding and predicting complex environments are essential for correct and safe operation. However, the deployment of such models in real systems requires careful consideration for ethical and safety challenges, such as biases in the training data or lack of transparency in the decision-making process.

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

# A. Limitations & Future Work

We recognize two main limitations that currently limit the scope of our PlaySlot framework to simple tabletop robotic scenarios, preventing it from generalizing to more complex domains.

**Limited Decomposition Model**  The first limitation arises from the SAVi object-centric decomposition model used at the core of our framework. SAVi achieves a great decomposition performance on datasets with objects of simple shapes and textures. However, it fails to generalize to complex real-world robotic scenarios, hence limiting the current scope of PlaySlot to robotic tabletop simulations, or simple real-world environments as in Sketchy.

**Single Latent Action**  The second limitation lies at the representational capability of our latent actions and action prototypes. In robotic scenarios, several actions often happen simultaneously, such as moving and rotating the robot, as well as opening or closing the gripper. Our current latent action representation jointly models the scene's inverse dynamics, encoding together all actions into a single latent space. This entangled representation limits our ability to control the agents in the scene with greater precision.

**Future Work**  In future work, we plan to extend our proposed PlaySlot framework with more capable decomposition models, such as DINOSAUR (Seitzer et al., 2023) or SOLV (Aydemir et al., 2023), as well as scale our inverse dynamics and predictor models. Furthermore, we can employ factorized latent action vectors, which represent in a disentangled manner different actions that happen simultaneously, such as moving the robot arm and opening the gripper. We believe that this architectural modifications will enable us to use PlaySlot on more complex robotic simulations and perform real-world robotic experiments.

# B. Implementation Details

In this section, we describe the network architecture and training details for each of the components in our PlaySlot framework. Our models are implemented in PyTorch (Paszke et al., 2017) and are trained on a single NVIDIA A100 GPU.

## B.1. Object-Centric Learning

We closely follow Kipf et al. (2022) for the implementation of the SAVi object-centric decomposition model, which we employ as scene parsing and object rendering modules. We strictly adhere to the architecture of their proposed CNN-based image encoder $\mathcal{E}_{\mathrm{SAVi}}$, slot decoder $\mathcal{D}_{\mathrm{SAVi}}$, transformer-based dynamics transition module, and Slot Attention corrector. We use a variable number of 128-dimensional object slots, depending of the dataset. Namely, we employ eight object slots on BlockPush, four slots on ButtonPress, three slots on GridShapes, and eight slots on Sketchy. On all datasets, we sample the initial object slots $\mathbf{S}_0$ from a Gaussian distribution with learned mean and covariance. Furthermore, we use three Slot Attention iterations for the initial video frame to obtain a good initial object-centric decomposition, and a single iteration for subsequent frames, which suffices to recursively update the slot state given the newly observed image features.

## B.2. Inverse Dynamics Module

We propose two variants of our InvDyn module.

**InvDyn$_{\mathrm{S}}$:** InvDyn$_{\mathrm{S}}$ jointly processes the object slots from a single time step $\mathbf{S}_t$ along with an additional token [ACT] using a transformer encoder. We use a four-layer transformer encoder with a 256-dimensional tokens, four 64-dimensional heads and hidden dimension of 1024. This module aggregates information from the object slots into the [ACT] token, and outputs a single latent action $\hat{\mathbf{z}}_t$ that captures the agent's action, making it well-suited for single-agent environments.

**InvDyn$_{\mathrm{M}}$:** InvDyn$_{\mathrm{M}}$ independently processes each object slot with a shared MLP, thus generating $N_{\mathbf{S}}$ latent action embeddings, each representing the action of a specific object in the scene. This design makes InvDyn$_{\mathrm{M}}$ well-suited for environments with multiple moving agents. We employ a two-layer MLP, featuring a ReLU nonlinear activation and layer normalization.

As described in Sec. 3.2.2, the generated latent actions $\hat{\mathbf{z}}_t$ are vector-quantized to assign them to their corresponding action prototype $\mathbf{p}_t$. On the ButtonPress, BlockPush, and Sketchy datasets, we use the InvDyn$_{\mathrm{S}}$ variant with eight different 16-dimensional action prototypes, whereas for GridShapes we use InvDyn$_{\mathrm{M}}$ with five distinct eight-dimensional action

prototypes. Following common practice, we update the action prototypes using the exponential moving average updates of cluster assignment counts (Van Den Oord & Vinyals, 2017).

### B.3. Conditional Object-Centric Predictor

Our conditional object-centric predictor (cOCVP) is inspired by the transformer-based SlotFormer (Wu et al., 2023a) architecture. Our cOCVP module features four layers, 256-dimensional tokens, eight 64-dimensional attention heads, and hidden dimension of 1024.

To enable predictions conditioned on the inferred latent actions, cOCVP maps the action prototypes $\mathbf{p}_{1:t}$, variability embeddings $\mathbf{v}_{1:t}$ and object slots $\mathbf{S}_{1:t}$ into the token dimensionality. The projected object slots are then conditioned by adding them with the projected action prototype and variability embedding from the corresponding time step. Furthermore, following Wu et al. (2023a), we augment these representations with a temporal sinusoidal positional encoding, such that all tokens from the same time step receive the same encoding, thus preserving the inherent permutation equivariance of the objects.

### B.4. Policy Model and Action Decoder

The policy model $f_\pi$ follows a similar architecture to InvDyn$_S$. $f_\pi$ is a four layer transformer that jointly processes the objects slots from a single time step $\mathbf{S}_t$ and an additional token [ACT] in order to regress a latent action.

The action decoder is a three-layer MLP with a hidden dimension of 128 that maps latent action vectors into real-world actions.

### B.5. Training Details

**SAVi Training:** SAVi is trained for object-centric decomposition using the Adam optimizer (Kingma & Ba, 2015), a batch size of 64, sequences of length eight frames, and a base learning rate of $10^{-4}$, which is linearly warmed-up for the first 4000 steps, followed by cosine annealing for the remaining of the training process. Moreover, we clip the gradients to a maximum norm of 0.05.

**InvDyn and cOCVP Training:** We jointly train our InvDyn and cOCVP modules given a pretrained SAVi decomposition model. These modules are trained with the Adam optimizer (Kingma & Ba, 2015), batch size of 64, and a base learning rate of $2 \times 10^{-4}$, which decreases during training with a cosine annealing schedule. To stabilize the training, we clip the gradients to a maximum norm of 0.05. We set the loss weights to $\lambda_{\text{Img}} = 1$, $\lambda_{\text{Slot}} = 1$, and $\lambda_{\text{VQ}} = 0.25$.

$f_\pi$ **and** $\mathcal{D}_\mathbf{a}$ **Training:** We train the $f_\pi$ and $\mathcal{D}_a$ modules given pretrained and frozen SAVi, InvDyn and cOCVP modules. These modules are trained with the Adam optimizer (Kingma & Ba, 2015), batch size of 64, and a fixed learning rate of $2 \times 10^{-4}$.

## C. Dataset Details

### C.1. BlockPush

This environment, inspired by Li et al. (2020) and simulated using MuJoCo (Todorov et al., 2012), features a robot arm on a tabletop interacting with multiple uni-colored blocks. We use two different dataset variants.

The first variant consists of a robot controlled by a random exploration policy, moving in the environment with minimal meaningful object interactions. This easily simulated dataset includes 20,000 training sequences and 2,000 validation sequences. We use this dataset variant for training SAVi, as well as for jointly training our inverse dynamics and conditional predictor modules.

The second variant contains a smaller subset of expert demonstrations where the robot is tasked to push the block of distinct color to a target location marked with a red sphere. This task evaluates the capabilities of an agent to reason about object relations and model object collisions. We collect 5,000 expert demonstrations using a pretrained policy (Mosbach et al., 2025) with a success rate of $\approx 80\%$ for this pushing task. We split this dataset into 4,500 training sequences, which are used for training the policy model and action decoder; and 500 evaluation sequences that are used for benchmarking the prediction models.

## C.2. ButtonPress

This environment, based on MetaWorld (Yu et al., 2020) features a Sawyer robot arm tasked with pressing a red button. Unlike BlockPush, it involves a non-object-centric task with complex shapes and textures. As before, we use two different dataset variants.

The first variant contains 10,000 sequences, split into 9,000 training and 1,000 validation videos, with the robot controlled by a random exploration policy. We use this dataset variant to train SAVi, as well as our inverse dynamics and conditional predictor modules.

The second variant includes a small subset of expert demonstrations where the robot successfully presses the red button. We collect 1,000 expert demonstrations using an expert policy, from which 900 are used for training the policy model and action decoder, and 100 demonstrations are used for benchmarking the prediction models.

## C.3. GridShapes

This dataset features one or more simple 2D shapes moving in a grid-like pattern on top of a colored background. The shapes can be either a ball, triangle or square, and have a random color. These shapes can move up, down, left, right or remain still, and revert their motion when an image boundary is reached, thus emulating a bouncing effect. To introduce some stochasticity into the motion, the shapes randomly change direction with a predefined probability of 0.25. We train and evaluate the models on several variants of the GridShapes dataset featuring different number of objects, ranging from one single moving shape, to five objects moving independently in the same sequence. This simple dataset serves as benchmark to evaluate a model's ability to jointly predict the motion of multiple moving agents in the scene.

## C.4. Sketchy

Sketchy (Cabi et al., 2020) is a real-world robotics dataset featuring videos in which a robotic gripper interacts with diverse objects of different shape and color on a tabletop scenario. We use the human demonstration sequences provided in the dataset, resulting in a total of 394 sequences with an average length of 80 frames. For each sequence, we use the observations taken with the `front right` and `front left` cameras, and discard the cropped images. We resize the video frames to an image size of $64 \times 64$ and use 316 and 78 sequences for training and evaluation, respectively.

# D. Baselines

In our experiments, we compare our approach with different baseline models, including the stochastic and playable video prediction models SVG (Denton & Fergus, 2018) and CADDY (Menapace et al., 2021), as well as the object-centric video prediction models SlotFormer (Wu et al., 2023a) and OCVP (Villar-Corrales et al., 2023). To ensure a fair comparison, we balance the number of learnable parameters and compute requirements for all methods. Tab. 3 lists the number of learnable parameters in our proposed PlaySlot as well as for all baselines.

Table 3: Number of learnable parameters for PlaySlot and baselines.

| Model | Trainable Parameters (M) |
|---|---|
| SVG | 18.84 |
| CADDY | 9.34 |
| SlotFormer | 3.97 (2.96 Predictor + 1.01 SAVi) |
| OCVP | 4.86 (3.85 Predictor + 1.01 SAVi) |
| PlaySlot | 8.49 (3.19 InvDyn + 4.27 cOCVP + 1.01 SAVi) |

## D.1. SVG

SVG (Denton & Fergus, 2018) is a generative model for video prediction that captures both deterministic dynamics and stochastic variations in video sequences. It combines a variational autoencoder (VAE) with recurrent neural networks (RNNs) to model stochastic temporal dynamics. SVG represents a probabilistic framework, where the next frame $\hat{\mathbf{X}}_{t+1}$ is generated based on the previous frame $\mathbf{X}_t$ and a latent sample $\hat{\mathbf{z}}_t$ drawn from a latent distribution. For a fair comparison with PlaySlot, we use SVG's posterior module to infer latent vectors using future video frames, and use those latent vectors

to guide the video prediction process. In our experiments, we adapt the original implementation[1] and use an SVG variant with a learned prior, VGG-like encoder and decoders, and two recurrent predictor layers.

**Main Difference with PlaySlot:** SVG operates with holistic scene representations, whereas our proposed method employs a structured object-centric representation. Furthermore, SVG encodes the stochastic scene dynamics into a single continuous latent vector. In contrast, PlaySlot follows a hybrid approach in which the latent action vectors are composed of an action prototype and action variability embeddings, making the prediction process more controllable and interpretable.

### D.2. CADDY

CADDY (Menapace et al., 2022) is a recurrent encoder-decoder model designed for playable video generation, enabling user-controllable future video prediction. CADDY infers latent actions that encode the agent's actions between consecutive pairs of frames. These latent actions are parameterized with a discrete *one-hot action label*, which determines the high-level action taking place; and a high-dimensional *action variability embedding*, which describes the variability of each action and captures the possible non-determinism in the environment. We adapt the original implementation[2] for our experiments.

**Main Difference with PlaySlot:** CADDY operates with holistic scene representations, i.e. CNN features, whereas our proposed method employs a structured object-centric representation. Moreover, despite both methods using a hybrid parameterization of the latent actions, they differ in their implementation. CADDY learns a discrete one-hot action label by minimizing multiple regularization objectives, including an action matching loss and several Kullback–Leibler (KL) divergences losses. In contrast, PlaySlot employs a discrete set of high-dimensional action prototypes, which are learned via vector-quantization of the latent space. We empirically verify that both approaches achieve comparable performance. However, our vector quantization approach requires significantly fewer hyper-parameters and is easier to tune.

### D.3. SlotFormer

SlotFormer (Wu et al., 2023a) is an object-centric video prediction model that builds upon slot-based representations. First, it uses Slot Attention to parse videos frames into object-centric latent representations, called slots. SlotFormer then employs a transformer-based autoregressive predictor module, which jointly processes all input slots in order to forecast future object representations. Finally, the predicted slots are decoded into object images and video frames. In our experiments, we adapt the original implementation[3].

**Main Difference with PlaySlot:** SlotFormer forecasts future slots in an unconditional manner, thus not being able to model stochastic environments or agents such as robots. PlaySlot addresses this challenge by inferring the scene inverse dynamics and using them to condition the prediction process.

### D.4. OCVP

OCVP (Villar-Corrales et al., 2023), similar to SlotFormer, is a slot-based object-centric video prediction model. Differing from SlotFormer, OCVP leverages two specialized decoupled attention mechanisms, *relational* and *temporal* attention, which model the object interactions and dynamics, respectively. In our experiments, we use the OCVP-Seq variant with default settings adapted from original implementation[4].

**Main Difference with PlaySlot:** OCVP forecasts future slots in an unconditional manner, thus not being able to model stochastic environments or agents such as robots. PlaySlot addresses this challenge by inferring the scene inverse dynamics and using them to condition the prediction process.

---

[1]https://github.com/edenton/svg
[2]https://github.com/willi-menapace/PlayableVideoGeneration
[3]https://github.com/pairlab/SlotFormer
[4]https://github.com/AIS-Bonn/OCVP-object-centric-video-prediction

# E. Additional Results

## E.1. Effect of the Number of Actions

In Tab. 4 we evaluate on the BlockPush dataset multiple PlaySlot variants using a different number of learned action prototypes. We show that using eight learned action prototypes achieves the best video prediction performance, while learning a concise semantically meaningful set of action prototypes.

Table 4: Evaluation of PlaySlot variants on the BlockPush dataset using a different number of learned action prototypes.

|  | BlockPush | | |
| --- | --- | --- | --- |
| # Actions | PSNR↑ | SSIM↑ | LPIPS↓ |
| 5 | 21.26 | 0.886 | 0.071 |
| 8 | **21.41** | **0.890** | **0.066** |
| 10 | 21.36 | 0.889 | 0.067 |
| 15 | 21.38 | 0.889 | 0.067 |

## E.2. Learned Behaviors from Expert Demonstrations

In this section, we present additional experiments and results demonstrating PlaySlot's ability to learn robot behaviors from unlabeled expert demonstrations. In Sec. E.2.1, we compare policies learned with PlaySlot under two different evaluation protocols. In Sec. E.2.2, we compare PlaySlot with LAPO (Schmidt & Jiang, 2024), a recent model for behavior learning from unlabeled demonstrations. Finally, Sec. E.2.3 provides additional rollout visualizations from PlaySlot's policy, along with simulated executions of the predicted action sequences.

### E.2.1. IMAGINED ROLLOUT VS. SIMULATION

In Fig. 9, we compare two different protocols for evaluating the behaviors learned with PlaySlot on the ButtonPress and BlockPush environments. Namely:

**w/ Latent Imagination** The learned behavior unfolds within the model's latent imagination, where future states are predicted with PlaySlot. To assess the performance of the policy, the predicted sequence of latent actions is decoded and executed in the simulator, with the final outcome—success or failure—used as the performance measure. This protocol assesses how well the learned representations and policies generalize within the internal model's constraints.

**w/ Simulation** The latent actions output by the policy are directly decoded and executed in the simulator, which generates the subsequent environment state. This process is repeated iteratively for a fixed number of steps or until the task is successfully completed. This protocol evaluates the quality of both the learned policy and action representation independently of the world model's accuracy, providing a more direct measure of the policy's effectiveness in an interactive setting.

As shown in Fig. 9, success rates improve for both evaluation protocols as the number of available expert demonstrations increases. Notably, with as few as 500 and 2000 expert demonstrations, PlaySlot w/ Simulator matches—and even

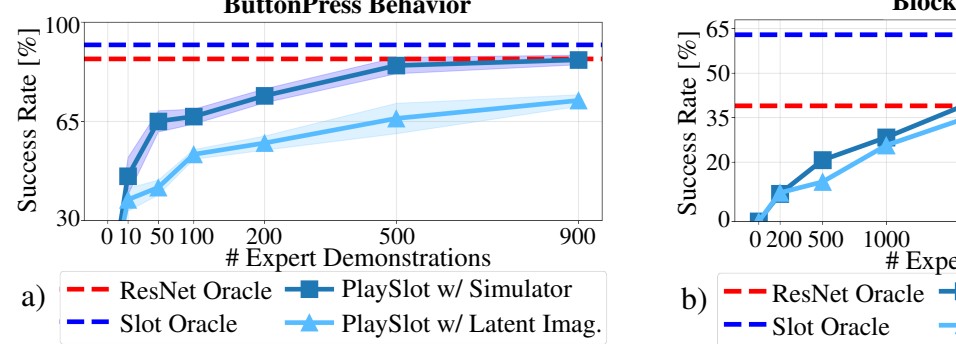

Figure 9: Success rates (%) of policies learned with PlaySlot under two evaluation protocols on the a) ButtonPress and b) BlockPush environments. Simulation-based rollouts outperform latent imagination, but both approaches improve with more available expert demonstrations and surpass the ResNet Oracle.

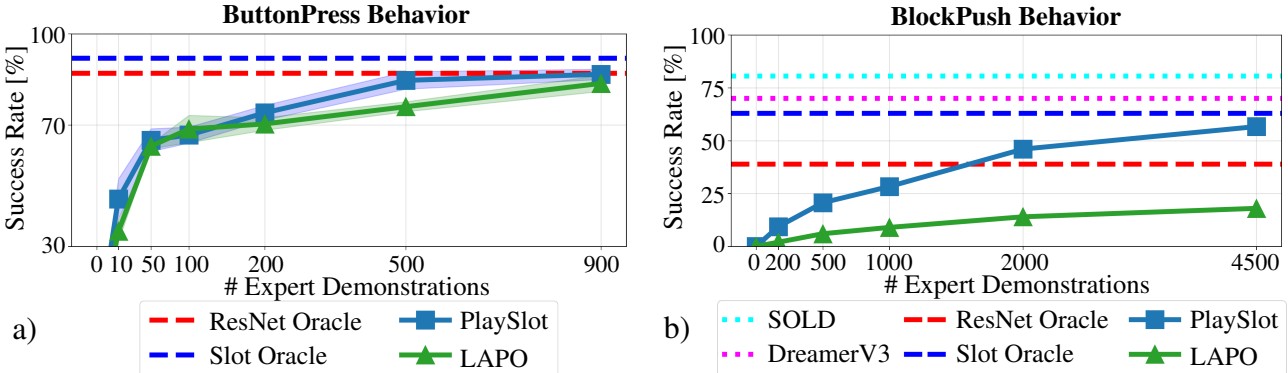

Figure 10: Success rate (%) as a function of available expert demonstrations on the a) ButtonPress and b) BlockPush environments. PlaySlot outperforms LAPO, especially on the challenging BlockPush environment, which requires reasoning about object properties.

outperforms—the ResNet Oracle baseline on the ButtonPress and BlockPush environments, respectively. This suggests that PlaySlot can effectively learn robust and performant robot behaviors from limited expert data when evaluated in an interactive setting.

Unsurprisingly, the latent imagination-based rollouts perform slightly worse than those executed in the simulator. This discrepancy arises because errors in the learned world model can compound over time, causing deviations from real-world dynamics. Nonetheless, the performance gap remains moderate, with the imagination-based rollouts succeeding approximately 70% and 50% of the scenes on the ButtonPress and BlockPush environments, respectively.

### E.2.2. COMPARISON WITH LAPO

LAPO (Schmidt & Jiang, 2024) is a recent model proposed for learning a world model, an inverse dynamics model, and a latent action policy from unlabeled videos. In contrast to the object-centric representations employed by PlaySlot, LAPO relies on feature maps output by a convolutional encoder.

**ButtonPress Behavior**   In Fig. 10a), we compare policies learned by PlaySlot and LAPO on the ButtonPress task, using a varying number of expert demonstrations. As reference baselines, we include two oracle models trained with access to all available expert demonstrations and to ground-truth actions: the Slot Oracle, which uses object-centric representations, and the ResNet Oracle, which uses ResNet feature maps.

Both PlaySlot and LAPO perform comparably on the ButtonPress task, which requires limited object-centric reasoning. Nonetheless, PlaySlot achieves slightly better sample-efficiency and consistently higher performance than LAPO across most data regimes. Notably, both methods approach the performance of the Oracle models (which have access to ground-truth actions during training) with a limited number of expert demonstrations. This is especially notable for PlaySlot, which matches the performance of the ResNet oracle with as few as 500 demonstrations.

**BlockPush Behavior**   On the challenging BlockPush task, which requires reasoning over object properties and relations, both models learn policies from a varying number of expert demonstrations, with only ≈80% depicting successful task completions—adding further difficulty due to noisy supervision.

As reference baselines, we include two model-based reinforcement learning methods, using holistic (DreamerV3 (Hafner et al., 2023)) and object-centric (SOLD (Mosbach et al., 2025)) latent spaces, both of which learn the robot behavior by directly interacting with the environment and using action and reward information. Additionally, we include two oracle behavior cloning baselines trained on all expert demonstrations, with access to ground-truth actions, and leveraging object-centric representations (Slot Oracle) or ResNet feature maps (ResNet Oracle).

Fig. 10b) shows that PlaySlot consistently outperforms LAPO by a large margin across all data regimes, demonstrating much stronger sample-efficiency and substantially higher final performance. While LAPO struggles to scale with the number of demonstrations, PlaySlot shows steady improvement, narrowing the gap between latent action models and baseline methods trained with access to ground-truth actions. Notably, PlaySlot even outperforms the ResNet Oracle model with as few as 2000 noisy expert demonstrations, and closely approaches the performance of the strong Slot Oracle baseline.

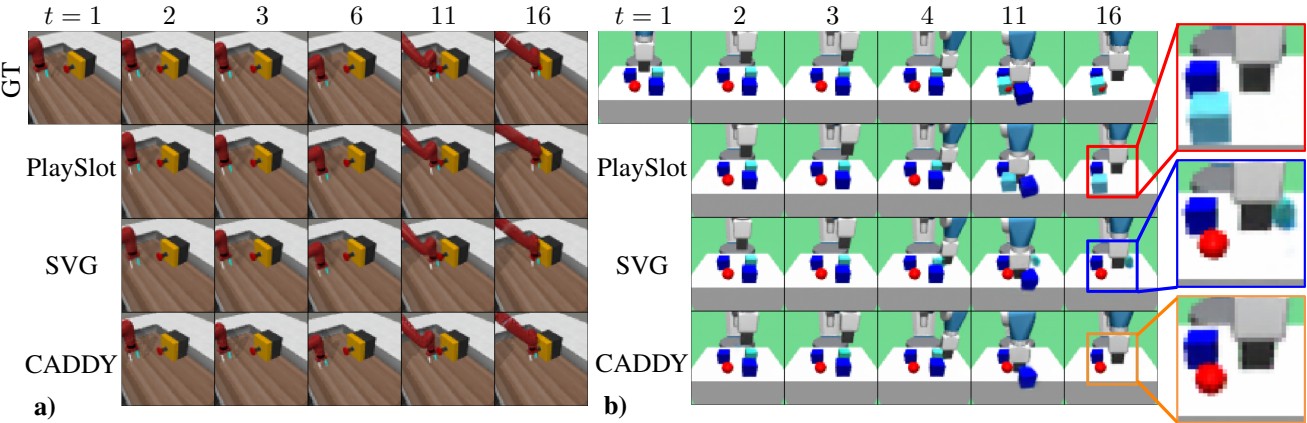

Figure 11: Qualitative comparison on (a) ButtonPress and (b) BlockPush datasets. Our method accurately predicts the scene dynamics, whereas the baselines fail to model object physics and interactions, leading to disappearing objects and failing to predict the pushing of the cyan block to the target location.

Additionally, we observe that slot-based object-centric models outperform their holistic counterparts, i.e. SOLD outperforms DreamerV3, Slot Oracle outperforms ResNet Oracle, and PlaySlot outperforms LAPO; further emphasizing the advantage of structured slot-based latent spaces.

These results highlight the benefits of object-centric representations for sample-efficient behavior learning, especially in tasks that require understanding object properties and their relations. By parsing the scene into individual objects, PlaySlot is able to generalize more effectively from limited and noisy demonstrations and infer complex behaviors, which are often challenging for models relying on monolithic, holistic representations.

### E.2.3. VISUALIZATIONS OF LEARNED BEHAVIORS

**ButtonPress Behavior:** Fig. 12 illustrates PlaySlot's learned behavior in the ButtonPress environment. The top row in each sequence (labeled as *Latent Behavior*) shows predicted trajectories within PlaySlot's latent imagination, where the model autoregressively generates actions using the policy model and predicts future scene states in the latent space. The bottom row (labeled *Sim. Actions*) depicts the simulated execution of the decoded latent actions in the environment. PlaySlot learns to solve the task within its latent imagination, successfully reasoning about the required action sequences before translating its latent actions into executable motions. Fig. 12b) shows a failure case where PlaySlot predicts within its latent imagination a trajectory that leads to successfully pressing the button. However, accumulated errors in the world model and during action decoding cause the simulated execution to miss the button.

**BlockPush Behavior:** Fig. 13 illustrates PlaySlot's learned behavior in the BlockPush environment. The top row in each sequence (labeled as *Latent Behavior*) shows predicted trajectories within PlaySlot's latent imagination, where the model autoregressively generates actions using the policy model and predicts future scene states in the latent space. The bottom row (labeled *Sim. Actions*) depicts the simulated execution of the decoded latent actions in the environment. PlaySlot learns to solve the task within its latent imagination, successfully reasoning about object properties and generating a precise sequence of latent actions, which can be decoded into executable motions. Fig. 13c) shows a failure case where PlaySlot controls the robot to interact with the correct block, but fails to place it in the target location.

### E.3. Qualitative Results

In the following sections, we provide additional qualitative results on all datasets, as well as qualitative comparisons with baselines. Further qualitative results and animations are provided in the project website.

### E.3.1. COMPARISON WITH BASELINES

Fig. 11 depicts a qualitative comparison between SVG, CADDY and PlaySlot on the ButtonPress and BlockPush datasets, respectively. On the ButtonPress dataset, as shown in Fig. 11a), all methods accurately model the trajectory of the robot arm. However, on the complex BlockPush task, depicted in Fig. 11b), SVG and CADDY fail to model the object collisions,

failing to move the block of distinct color to the target location, and leading to blurriness and disappearing objects. In contrast, PlaySlot maintains sharp object representations and correctly models interactions between objects, leading to accurate frame predictions.

### E.3.2. BLOCKPUSH DATASET

Fig. 14 shows two qualitative comparisons between PlaySlot, CADDY and SVG on the BlockPush dataset. Our proposed method, which explicitly models object interactions, preserves sharp object representations and accurately predicts future frames. In contrast, SVG and CADDY, which rely on holistic scene features for forecasting, struggle to model object collisions, resulting in blurry predictions and disappearing objects.

Fig. 15 shows the predicted video frames, slot masks, and objects representations on a BlockPush sequence. PlaySlot parses the scene into precise object images and masks, which can be assigned a unique color to obtain a segmentation of the scene. Our approach decomposes the BlockPush environment using eight object slots, where one slot represents the background, five slots to different blocks, one slot to the red target, and one slot to the robot arm. The sharp object images and masks demonstrate that PlaySlot encodes into each slot features from the corresponding object. This allows our method to directly reason about object properties, dynamics and interactions, allowing for accurate future frame predictions.

Fig. 16 depicts the effect each action prototype learned by PlaySlot on the BlockPush dataset. PlaySlot learns consistent semantically meaningful action prototypes that control the robot arm to move in a specific direction. We note that some actions prototypes, e.g, action 5 and 7, perform semantically similar actions but with different velocities.

### E.3.3. BUTTONPRESS DATASET

Fig. 17 shows a comparisons between PlaySlot, CADDY and SVG on the ButtonPress dataset. All methods successfully predict the ground truth sequence by inferring and modeling the robot's dynamics,

Fig. 18 shows PlaySlot's predictions and object representations on a ButtonPress sequence. We visualize the ground truth sequence, the predicted frames, segmentation obtained by assigning a different color to each slot mask, as well as the object reconstructions for four slots. PlaySlot assigns one slot to the background, one slot for box and red button, and two slots for different parts of the robot arm.

Fig. 19 depicts the effect each action prototype learned by PlaySloton the ButtonPress dataset.

### E.3.4. GRIDSHAPES DATASET

Fig. 20 depicts the effect each action prototype learned by PlaySlot on a variant of GridShapes with three shapes. PlaySlot successfully captures the five possible object movements—up, down, left, right, and stay—predicting future frames by modeling the motion of each object independently. However, we observe that PlaySlot sometimes generates artifacts when shapes reach the image boundaries. We attribute this to the training data, where objects change direction upon reaching the boundary mimicking a bouncing effect. This leads to poor prediction performance when conditioning the model to predict outside its training distribution.

### E.3.5. SKETCHY DATASET

Fig. 21 shows PlaySlot's predictions and object representations on a Sketchy sequence. We visualize the ground truth sequence, the predicted frames, segmentation obtained by assigning a different color to each slot mask, as well as the object reconstructions for four slots. PlaySlot assigns two slots to the workspace and background, two slots for each part of the robot gripper, and two slots for different objects present in the scene.

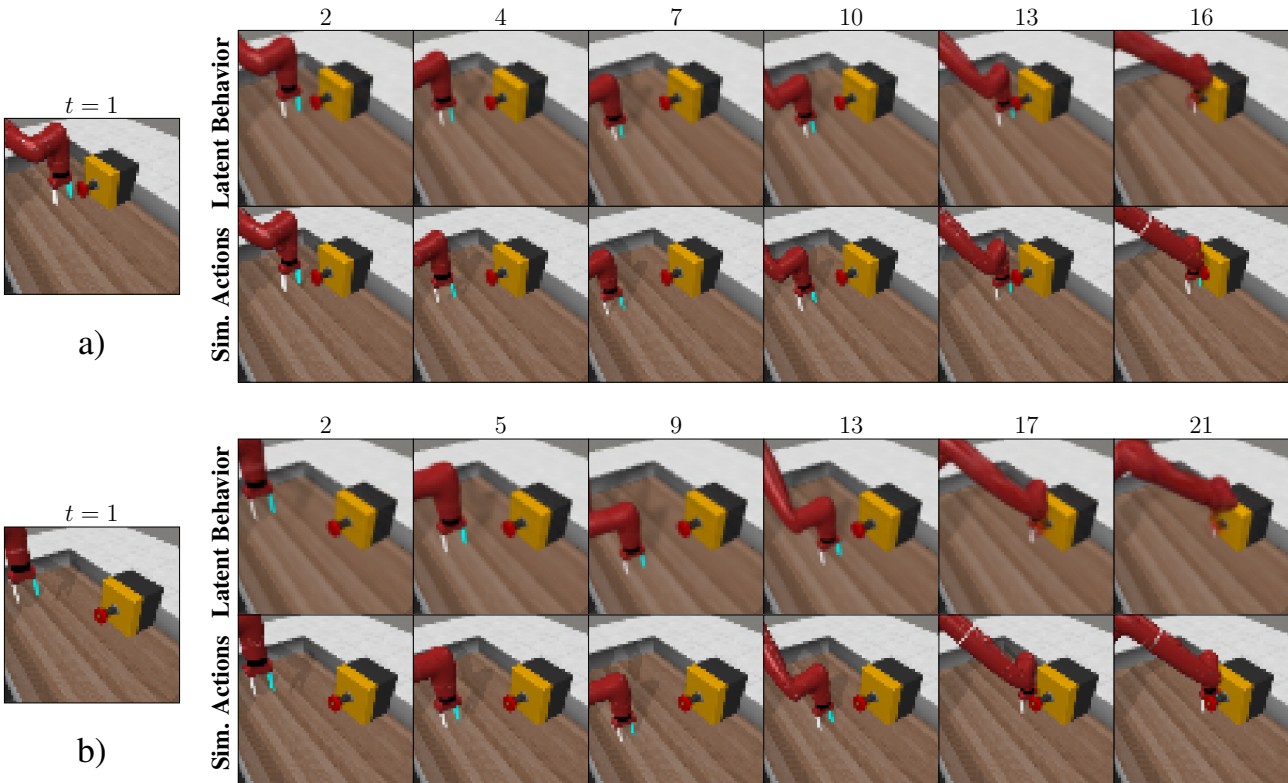

Figure 12: Predicted frames using latent actions from the learned policy, and sequences simulated by executing the decoded latent actions for two ButtonPress scenarios. **a)** PlaySlot successfully plans the trajectory within its latent imagination and translates latent actions into executable motions. **b)** PlaySlot fails to decode its latent actions into correct robot motion, missing the button.

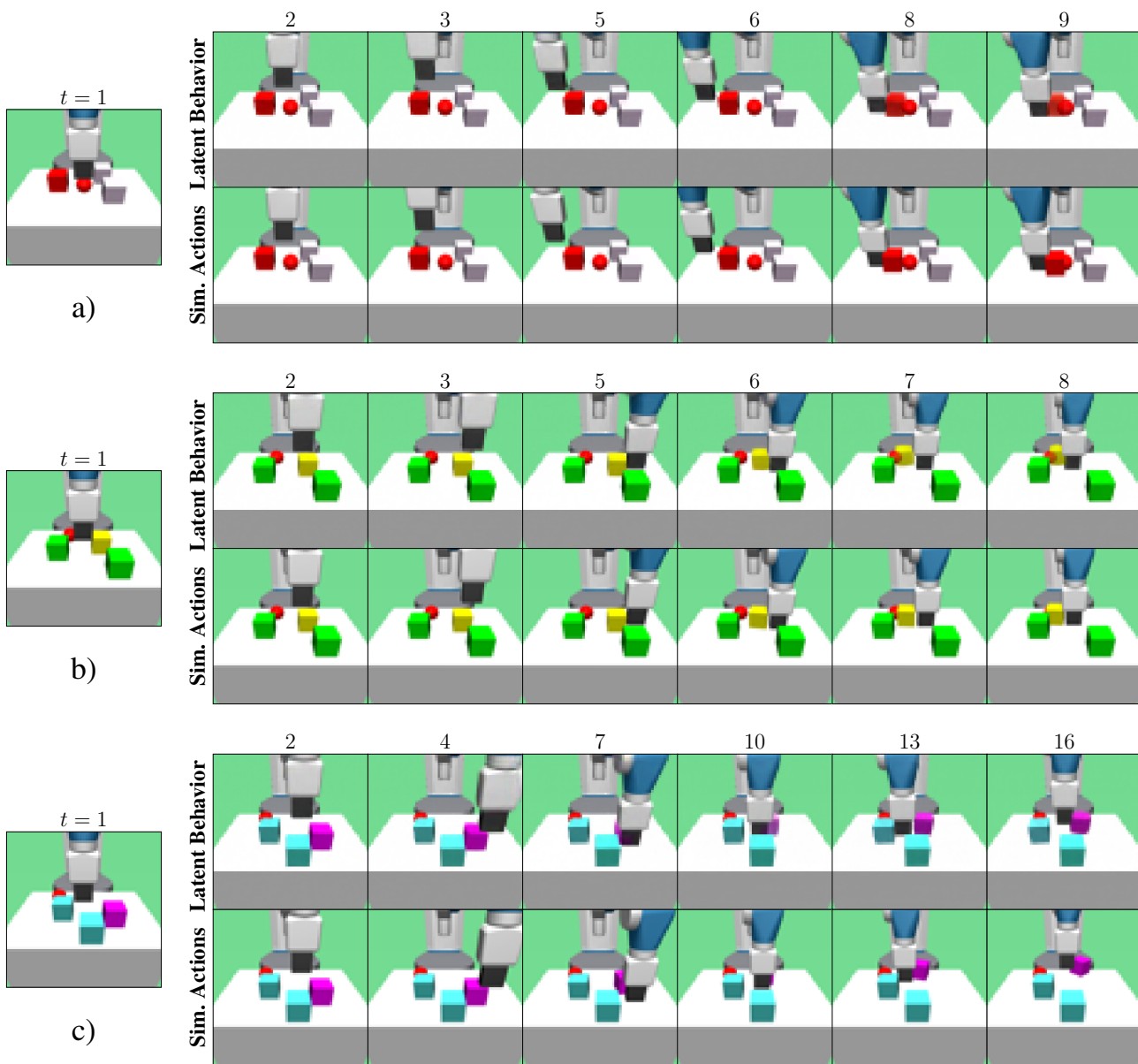

Figure 13: Predicted frames using latent actions from the learned policy, and simulation computed by executing the decoded latent actions for three BlockPush scenarios. **a) & b)** PlaySlot identifies the block of distinct color and generates executable latent actions to push it to the target location. **c)** PlaySlot identifies the block but fails to push it into the target location.

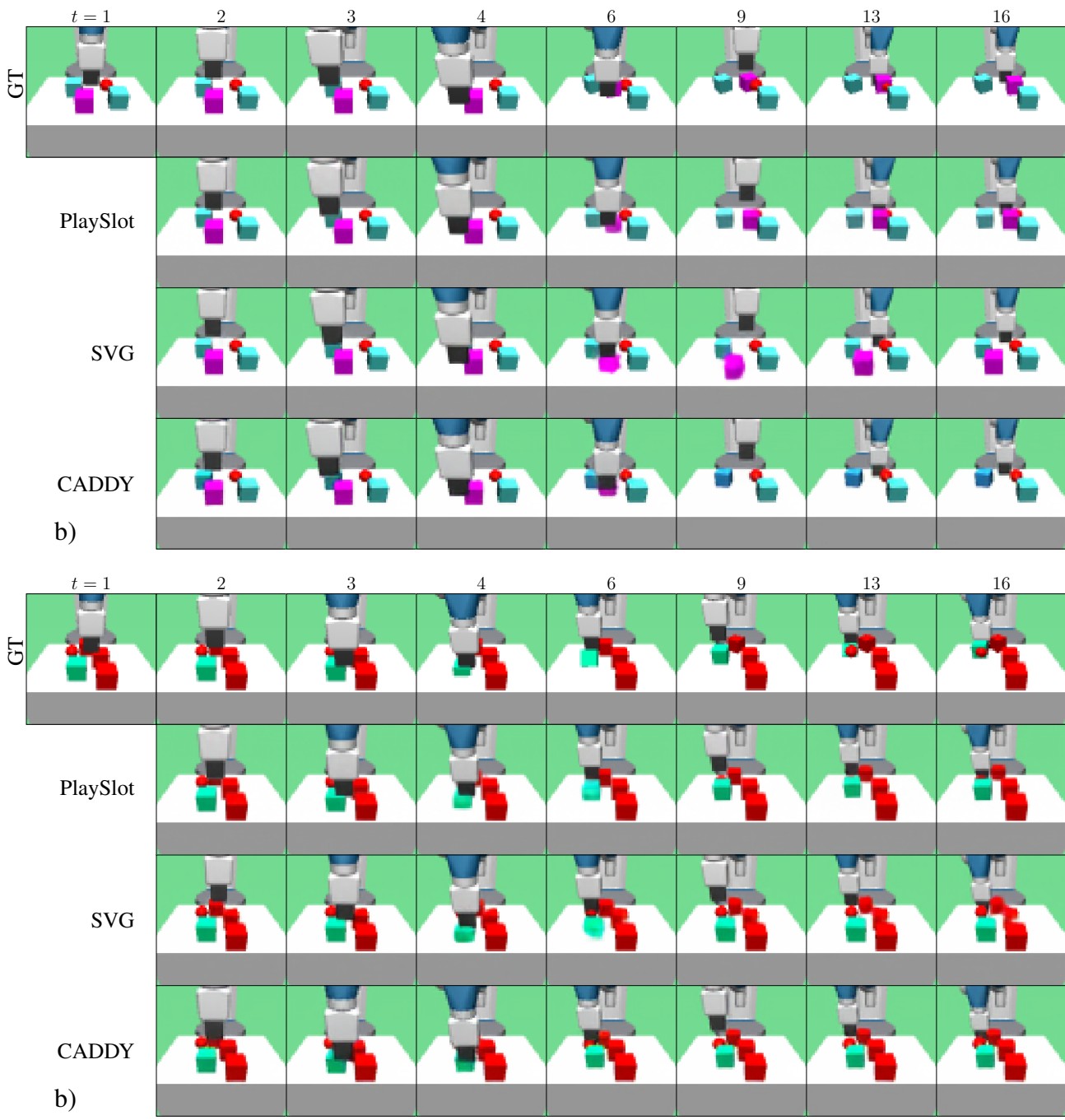

Figure 14: Qualitative comparison on BlockPush. Our method accurately predicts the scene dynamics and object interactions, whereas the baselines fail to model object collisions, leading to blurry or disappearing objects.

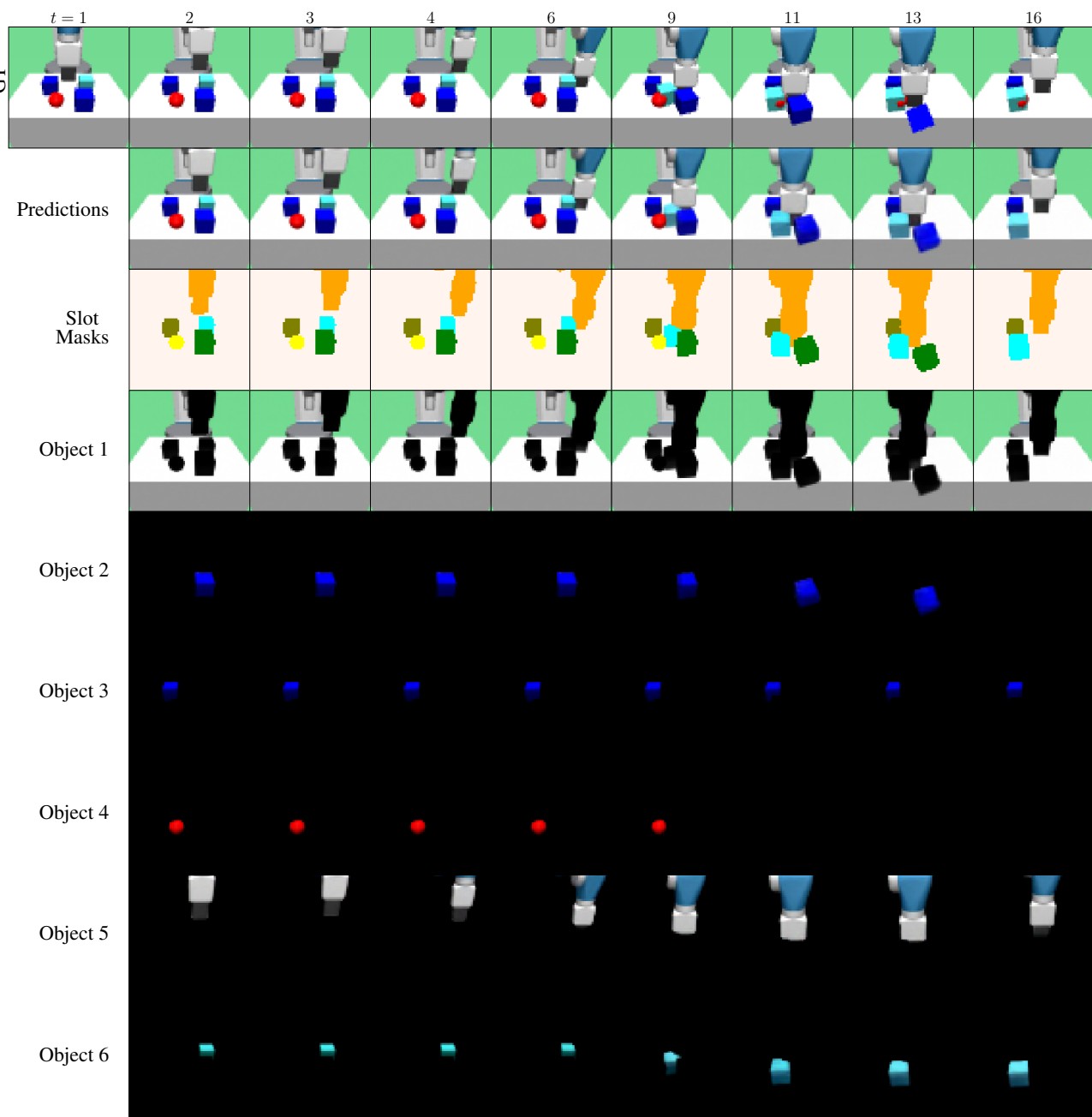

Figure 15: PlaySlot predictions and object representations on a BlockPush sequence. We visualize the ground truth sequence, the predicted frames, segmentation obtained by assigning a different color to each slot mask, as well as the object reconstructions for five slots. PlaySlot assigns one slot to the background, one slot for the robot, one slot for the red target, and the remaining slots to the blocks.

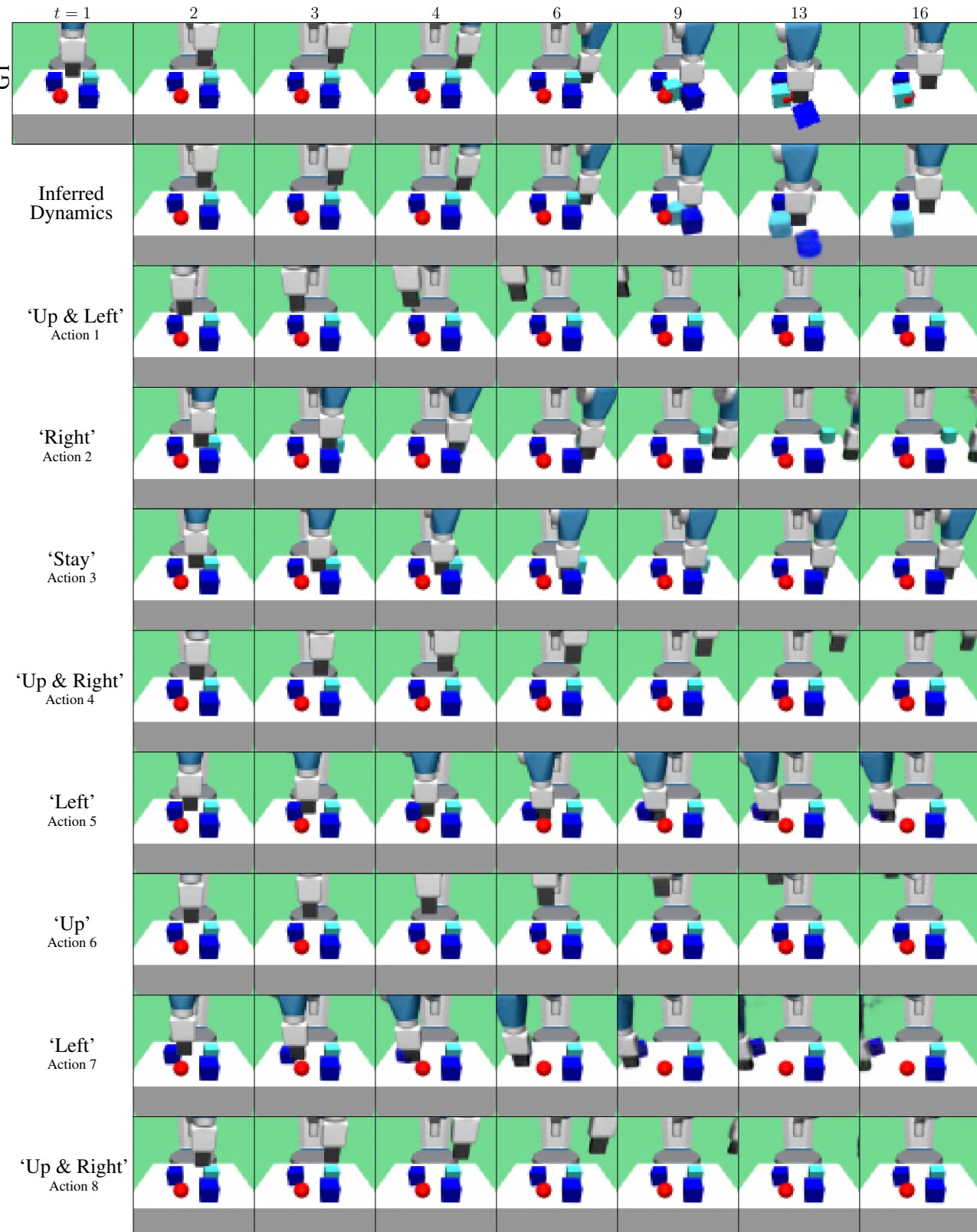

Figure 16: PlaySlot predictions conditioned on different latent actions, including the inferred inverse dynamics, as well as each action prototype learned on BlockPush. We generate a sequence by repeatedly conditioning the prediction process on a a single action prototype. The model learns action prototypes that control the robot to move consistently on a specific direction.

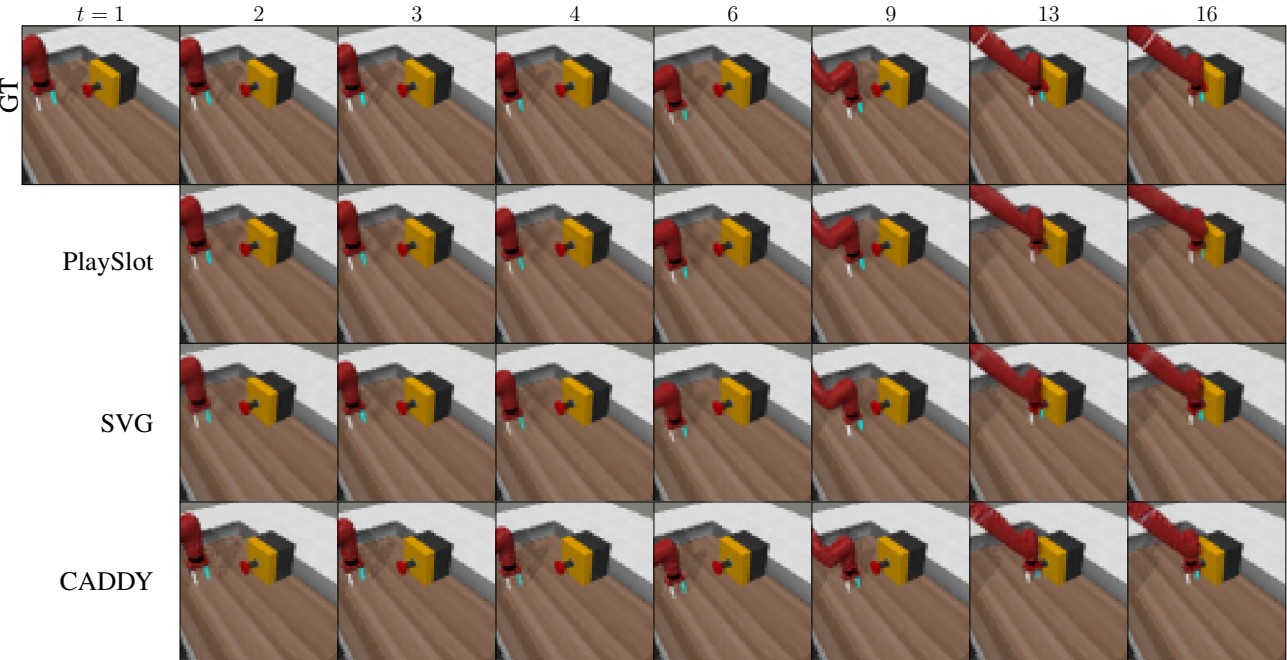

Figure 17: Qualitative comparison on ButtonPress. All methods successfully reconstruct the ground truth sequence by inferring the robot's trajectory.

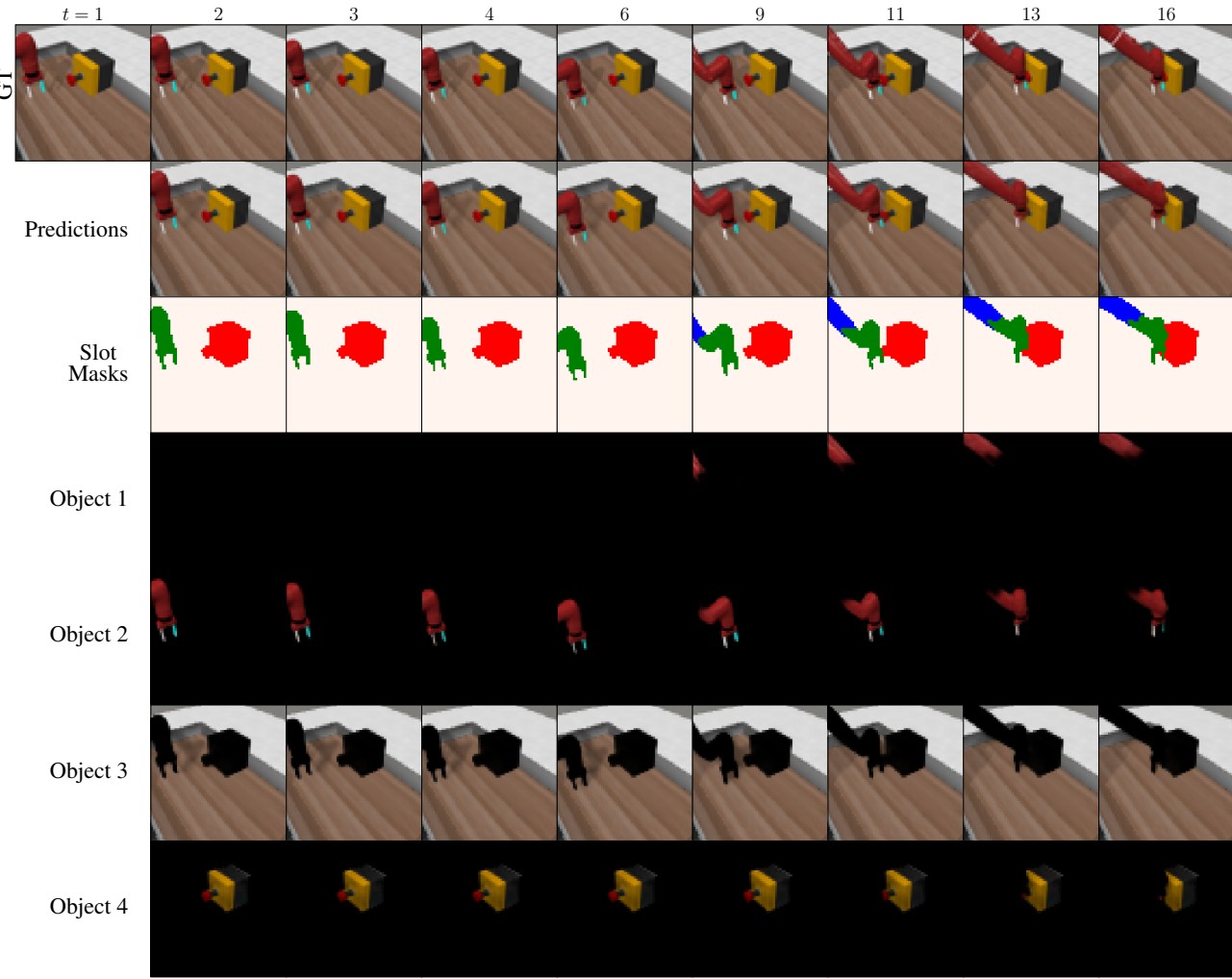

Figure 18: PlaySlot predictions and object representations on a ButtonPress sequence. We visualize the ground truth sequence, the predicted frames, segmentation obtained by assigning a different color to each slot mask, as well as the object reconstructions for four slots. PlaySlotassigns one slot to the background, one slot for box and red button, and two slots for different parts of the robot arm.

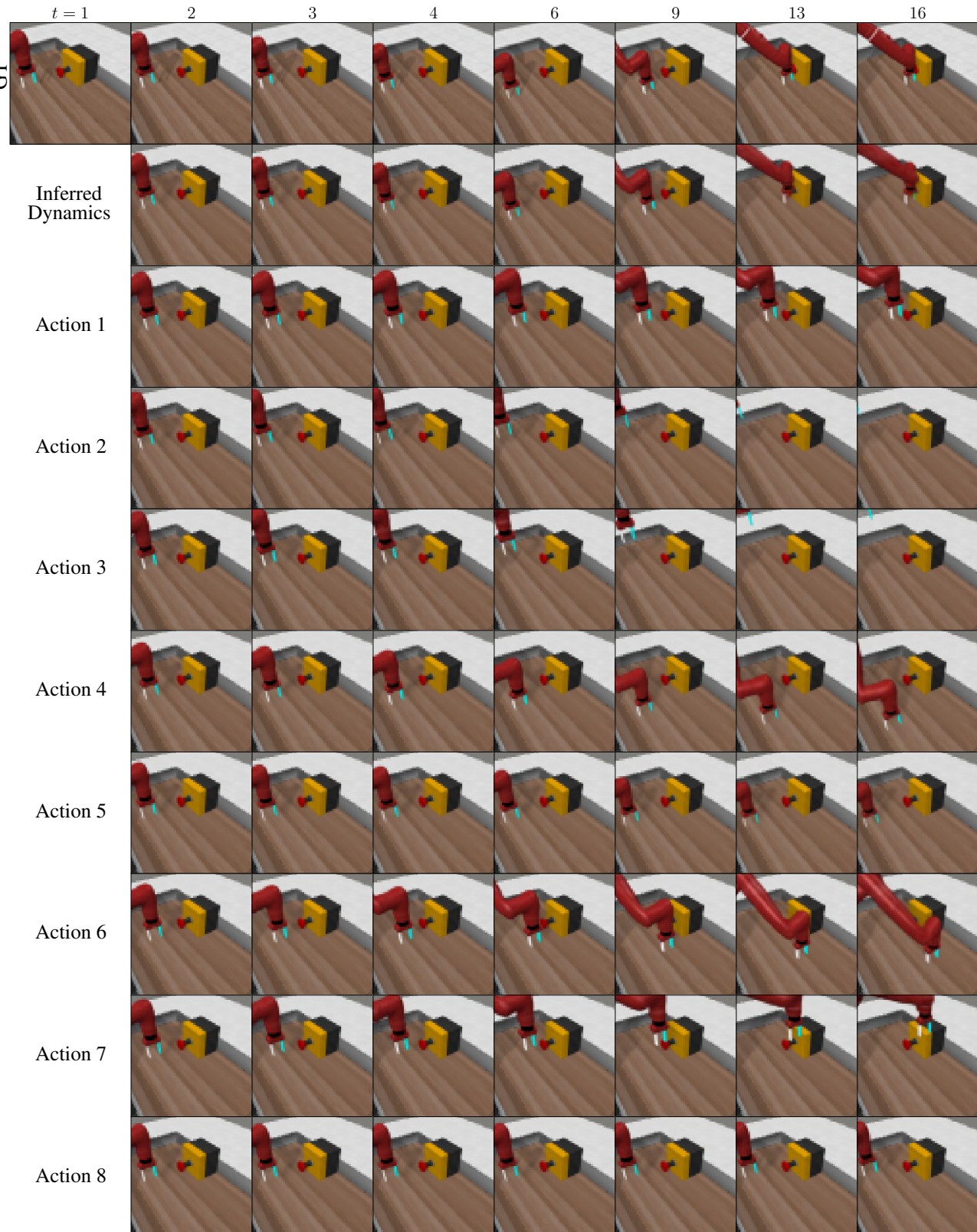

Figure 19: PlaySlot predictions conditioned on different latent actions, including the inferred inverse dynamics, as well as each action prototypes learned on ButtonPress. We generate a sequence by repeatedly conditioning the prediction process on a a single action prototype. The model learns action prototypes that control the robot to move consistently on a specific direction.

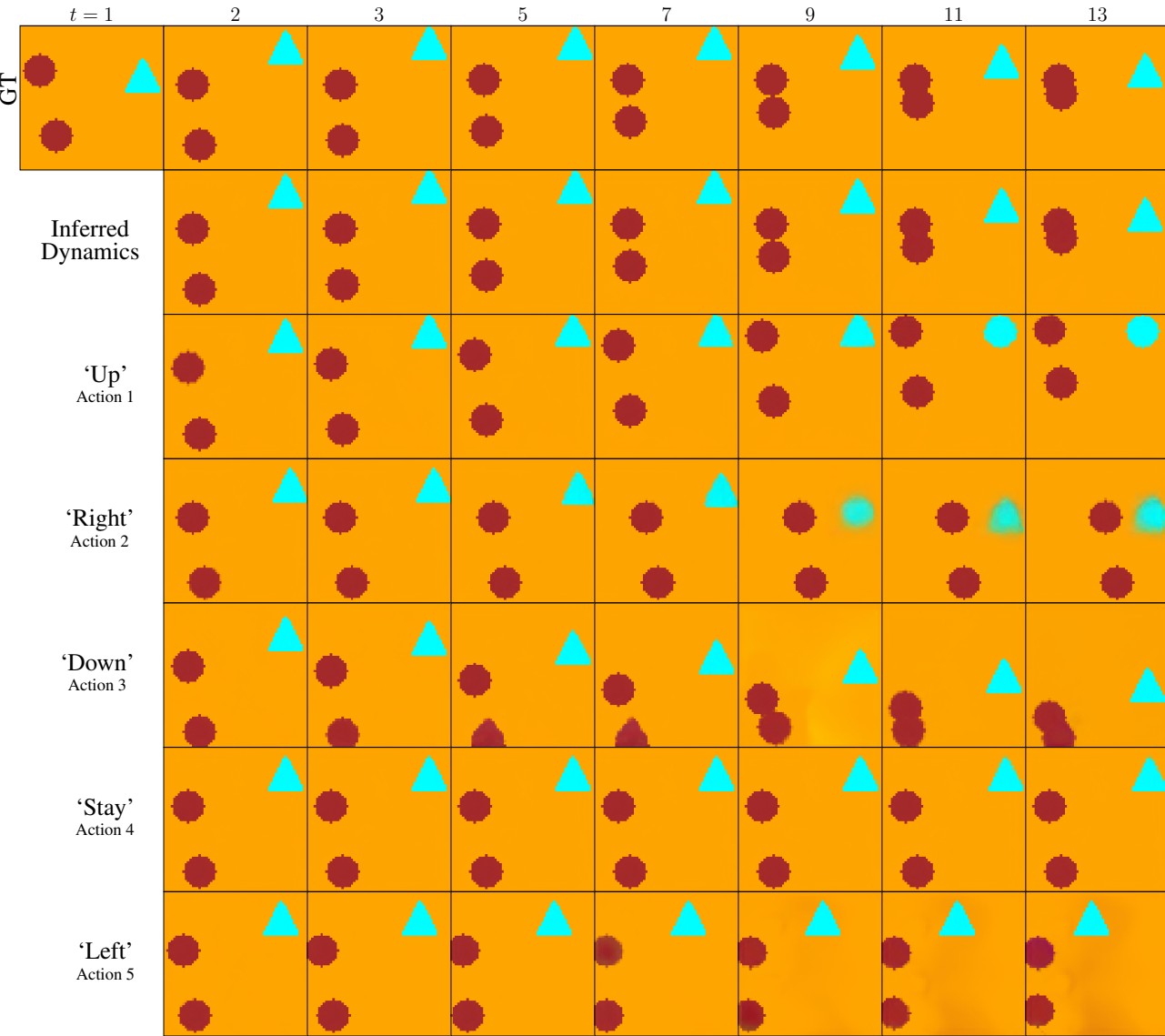

Figure 20: PlaySlot predictions conditioned on different latent actions, including the inferred inverse dynamics, as well as each action prototypes learned on the GridShapes dataset. We generate a sequence by repeatedly conditioning the prediction process on a a single action prototype. The model learns the five possible actions and achieves sharp predictions by forecasting the motion of each object individually.

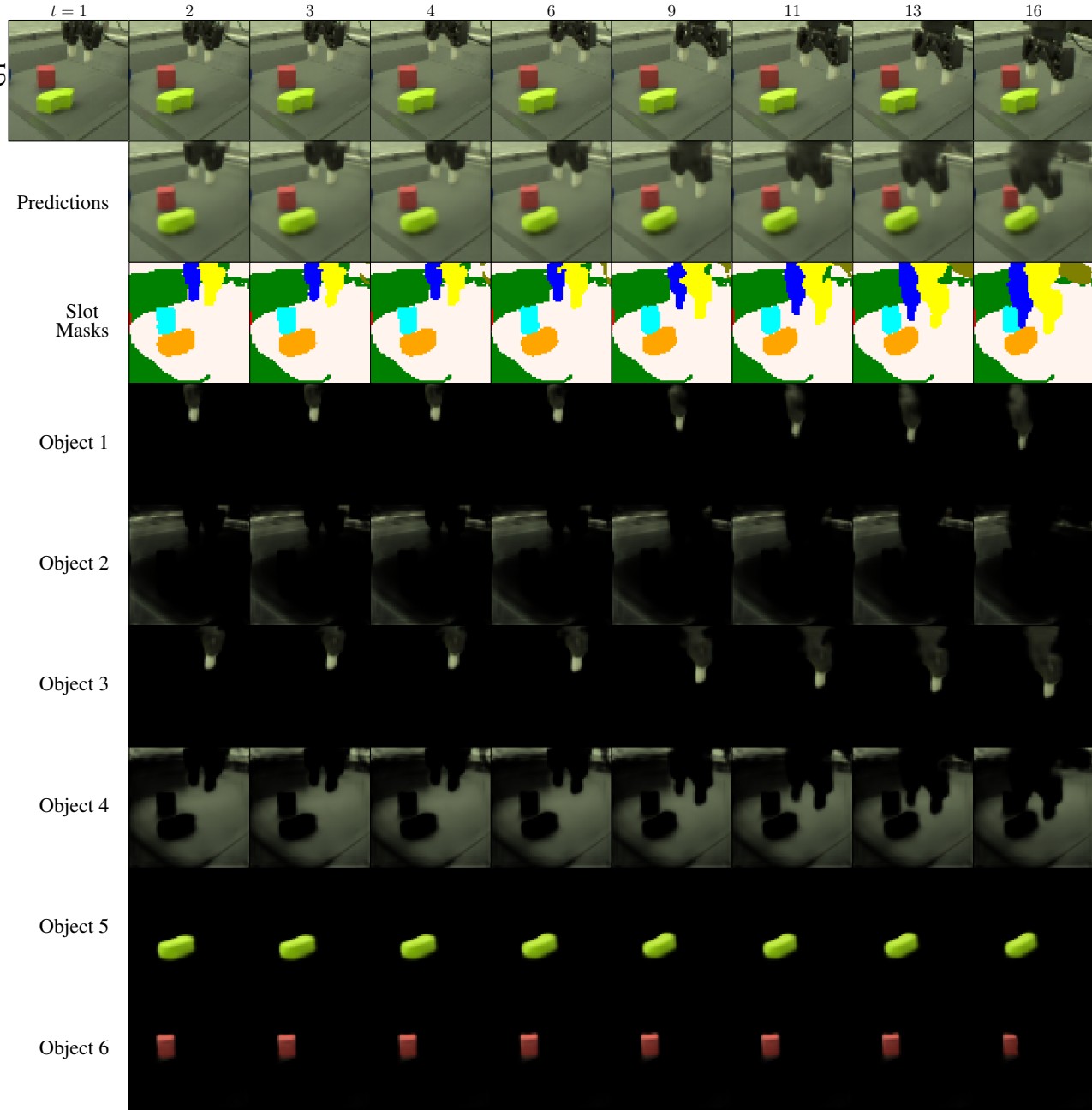

Figure 21: PlaySlot predictions and object representations on a Sketchy sequence. We visualize the ground truth sequence, the predicted frames, segmentation obtained by assigning a different color to each slot mask, as well as the object reconstructions for four slots. PlaySlot assigns two slots to the workspace and background, two slots for each part of the robot gripper, and two slots for different objects present in the scene.

