# OpenReview forum: "PlaySlot: Learning Inverse Latent Dynamics for Controllable Object-Centric Video Prediction and Planning"
_ICML.cc/2025/Conference — ICML 2025 poster_

### Official Review · Reviewer_nim7 · 2025-02-20

**Overall Recommendation:** 2

**Summary:**

The paper tackles the problem of learning controllable object-centric video prediction from actionless videos. The proposed method, PlaySlot, combines slot-attention-based object-centric representations with a previous slot dynamics prediction module (OCVP) conditioned on learned latent actions (inverse module). A latent policy is also learned to infer latent actions at inference time. The model is evaluated on robotic datasets, two simulated and one real-world (Sketchy).

**Claims And Evidence:**

* Claims are clear – designing a controllable video prediction model.
* Evidence: very marginal improvement over baselines (except the simplistic GridShapes dataset), especially over the “holistic” baselines (like CADDY). It doesn’t seem that the object-centric decomposition provides significant gains, except for certain cases where the “holistic” approaches don’t model object-interactions, but these cases seem rare as evident by the close pixel-metrics (also, hard to tell without video rollouts of the baselines how severe the issue is).
* Evidence: no quantitative results on the real-world Sketchy dataset or comparison with baselines on that dataset.
* Evidence: in the provided video rollouts, the predictions (e.g., on Sketchy) don’t seem very “stable”, objects randomly move/wiggle There are no video rollouts of the baselines.

**Essential References Not Discussed:**

The paragraph “Unsupervised Object-Centric Learning” under the Related Work section should be renamed to “Unsupervised Slot-based Object-centric Learning” as it completely ignores other unsupervised OC families, such as the patch-based family (e.g., SPACE-https://arxiv.org/abs/2001.02407) and the particle-based family (e.g., DLP-https://arxiv.org/abs/2205.15821). The other approaches don’t use the term “slots”, so if you decided to ignore the previous work in unsupervised OC, use the correct terms. While it is true that the object-centric community is mostly focused on slot-based models, I find it a bad habit to ignore the other types of object-centric models.

**Experimental Designs Or Analyses:**

* The experimental design is sound.
* There are ablations.
* I raised some questions under the questions section and had several concerns under “Claims and Evidence”.

**Methods And Evaluation Criteria:**

* The proposed method and idea make sense for the problem; however, the method is not entirely clear.
* The benchmarks make sense, but they are too simple to draw a conclusion regarding the performance of the proposed method.

**Other Comments Or Suggestions:**

Add a reference in the main text to limitations and future work in the appendix.

**Other Strengths And Weaknesses:**

**Strengths**:
* Open-source code.

**Weaknesses**:
* The paper is hard to follow and includes obscure descriptions of the components (especially regarding the latent actions modules).
* No video rollouts of the baselines.

**Questions For Authors:**

* In the provided video rollouts, the predictions (e.g., on Sketchy) don’t seem very “stable”, objects randomly move/wiggle. Is there an explanation for that? Maybe the latent action and slots are not entirely disentangled (I would expect static objects to remain static unless interacted with).
* Latent action design choices: to my understanding, the sampled Gaussian latent action is quantized? I find this an odd design choice. How is $v_t$, the “action variability embedding” modeled? This information is missing in Section 3.2.1 (or have I missed it?). According to lines 244-251, at inference, $v_t = \hat{z}_t - p_t$. Is that what happens at train time as well? $v_t$ is the residual?
* I thought the whole-point was to learn from actionless videos, but it seems that you regress the ground-truth real actions with an action-decoder? This information is obscured from the reader until Section 3.5. So does this approach require labeled actions? Or is it used for evaluation or task-related (Section 4.4) purposes only? It is unclear from the text how the latent actions are learned, and what is the use of the action decoder that requires labeled real actions?
* How does one choose between the inverse model variants? Is it just trial-and-error? Shouldn’t the M variant work-well for all cases?

**Relation To Broader Scientific Literature:**

* The key contribution of this paper is complimenting a slot-based object-centric video-prediction model with a latent action module, which enables handling stochastic transitions (e.g., an action performed by a robot).
* In that sense, there is a contribution to the community; however, the chosen benchmark includes simple simulated datasets and one real-world dataset (with no comparisons to baselines).
* The performance of the method is not significant with respect to the baselines.

**Theoretical Claims:**

There are no theoretical claims in this paper.

---

> ### Author Rebuttal · Authors · 2025-04-01
>
> Thanks for the constructive review. We are happy that you find our claims clear and our experimental design sound.
> For the final version, we will try to improve the description of the latent action modules and we will rename the subsection to “Unsupervised Slot-based Object-centric Learning”.
> Below we address your specific questions and comments:
>
> **Predictions on Sketchy don’t seem very “stable”. Is there an explanation for that?**
>
> This is indeed an interesting observation. We attribute this behavior to two main factors:
>  - *Simplicity of rendering module*: PlaySlot (following SAVi) uses a shared simple CNN to decode slots into object images and masks, which only interact with each other via a weighted sum. We hypothesize that a more expressive rendering module that jointly decodes all slots, e.g. transformer-based or diffusion-based, could overcome this limitation.
>
>  - *Error-Accumulation*: Our object-centric predictor is an autoregressive transformer, which can be susceptible to error accumulation over long prediction horizons. Since each predicted frame conditions the next, small inaccuracies in early steps can propagate and amplify, leading to noticeable object drift or flickering. One interesting future research direction would be to introduce temporal regularizations to minimize this effect.
>
> Nevertheless, we observe that baselines such as SVG or CADDY also suffer from lack of temporal consistency on Sketchy: https://anonymous.4open.science/r/Rollouts-7F68/
>
>
> **Is the sampled Gaussian latent action quantized? How is the $\mathbf{v}$ modeled? Is it the residual?**
>
> The latent actions $\mathbf{\hat{z}}$ are indeed sampled from a learned Gaussian distribution.
> Additionally, we decompose each latent action into an action prototype $\mathbf{p}$ and a variability embedding $\mathbf{v}$. This hybrid parameterization helps us learn semantically meaningful action prototypes, while also being able to model more complex actions (e.g. interpolations or modifications) by using the variability embedding.
> Namely:
>  - $\mathbf{p}$ is obtained by vector quantizing the latent action: $\textbf{p} = VQ(\mathbf{\hat{z}})$
>  - The variability $\mathbf{v}$ is the residual: $\mathbf{v} = \mathbf{\hat{z}} - \mathbf{p}$
>
> These specifics on how $\textbf{p}$ and $\textbf{v}$ are computed are described in Sec. 3.2.2 Lines 187-192-Col.2, but we will make this also explicit in Sec. 3.2.1.
>
>
> **Use of real actions to train the action-decoder:**
>
> PlaySlot learns to infer latent actions and learns robot behaviors in a completely unsupervised manner from actionless videos.
> An action decoder is trained to map latent actions to real-actions, but this module is only needed to execute latent actions in the simulato for evaluation. See detailed answer to reviewer 19cQ.
>
> We will try to make this clearer in the final version of the paper.
>
>
> **How does one choose between the inverse model variants?**
> If we know that there is a single agent in the scene (e.g. tabletop robotic scenarios), we use the InvDynS variant.
> When multiple objects (> 2) are moving, we show that using a single latent action is not powerful enough, and explicitly modelling the action of each agent/object using InvDynM proves to be beneficial.
> Therefore, we use InvDynS on the robotic scenarios, and InvDynM on GridShapes.
>
>
> **No quantitative results on the real-world Sketchy dataset or comparison with baselines on that dataset.**
> We now include a quantitative evaluation of PlaySlot and baselines on Sketchy.
> See detailed answer to review R5dN.
>
> Additionally, we provide videos in the following link featuring qualitative comparisons with the baselines on Sketchy and all other datasets: https://anonymous.4open.science/r/Rollouts-7F68/
>
>
> **Marginal improvement over baselines**
>
> While pixel-wise metrics are indeed close in some cases, our method still demonstrates notable advantages. In particular, on BlockPush, PlaySlot achieves the best results, highlighting its ability to model object dynamics and interactions effectively. Furthermore, PlaySlot achieves the overall best performance on Sketchy (see table in response to reviewer R5dN) and remains competitive in non-object-centric scenarios such as ButtonPress.
>
> It is important to note that BlockPush contains object interactions in every test scene, where the robot must push a block to a target location. Among the compared methods, only PlaySlot consistently captures these interactions and generates coherent sequence predictions.
>
> Moreover, while small object sizes in BlockPush may limit the impact of object localization errors on quantitative video metrics, this does not diminish the benefits of object-centric decomposition.
>
> Additionally, beyond immediate gains in pixel-based scores, our InvDyn module, in combination with a structured slot-based representation, enables sample-efficient behavior learning, as detailed in App. E.2. This makes PlaySlot particularly valuable in settings where data efficiency is crucial.

---

> > ### Comment · Reviewer_nim7 · 2025-04-03
> >
> > I appreciate the authors effort during the rebuttal period. The additional clarifications and evaluations indeed make the picture a bit more clearer. However, it seems that the choice between model variants requires priviledged knowledge of the environment and I'm still unconvinced by the quantitative results on Sketchy, seems like there is not much improvement (if any, seems like there is no statistical significance). The method is interesting but its performance is not entirely convincing. While most of the object-centric models are evaluated on synthetic datasets, I think it is worthwhile to demonstrate them on real-world datasets (this is why I truly appreciate the Sketchy experiments). Overall, I'm open to increasing my score after further discussion with the reviewers and AC, but currently I find it hard to do so as I don't think there is enough evidence the proposed approach indeed improves upon the baselines (and I acknowledge that the authors do not agree with the reviewers on this point). Thank you again for your effort.

---

> > > ### Author Response · Authors · 2025-04-08
> > >
> > > Thank you for engaging with our response and for taking the time to consider our clarifications and additional results. We truly appreciate your thoughtful comments and your openness to further discussion. We are especially glad to hear that you value our Sketchy experiments — we also believe that further pushing object-centric models towards real-world applications is an important and necessary step for the field.
> > >
> > > We understand your concerns regarding the performance improvements, especially on Sketchy. While we acknowledge that some of the gains may appear marginal in terms of pixel-level metrics, and that PlaySlot may not universally outperform all baseline methods, we believe that slot-based object-centric representations offer meaningful benefits for world modeling and behavior learning. These advantages include explicit modeling of object relations and interactions, improved interpretability, and a natural structured representation for downstream control applications — beneficial properties that are not always fully captured by standard quantitative metrics. That said, we appreciate your perspective and agree that stronger quantitative comparisons would help strengthen the case, and this is something we aim to improve upon in future work.
> > >
> > > To further highlight the strengths of PlaySlot and the use of a slot-based structured latent space, we have compared our proposed PlaySlot with LAPO (Schmidt & Jiang. ICLR. 2024),  a recent method for sample-efficient behavior learning, on both the *ButtonPress* and *BlockPush* environments.
> > > Specifically, we evaluated and compared how effectively each method learns target robot behaviors in a sample-efficient manner from a limited number of expert demonstrations.
> > > A detailed discussion of these experiments and results is provided in:
> > >  - https://anonymous.4open.science/r/Rollouts-7F68/BEHAVIOR_LEARNING.md
> > >
> > > Below, we summarize the key findings:
> > >
> > >  - PlaySlot and LAPO perform comparably on the *ButtonPress* environment, which is less reliant on object-centric reasoning.  Nonetheless, PlaySlot achieves slightly better sample-efficiency and higher performance than LAPO across most data regimes.
> > >
> > >  - On the more challenging *BlockPush* task — where understanding object properties and their relations is crucial — PlaySlot consistently outperforms LAPO by a large margin across all data regimes, demonstrating much stronger sample-efficiency and substantially higher performance.
> > >
> > > These results highlight the strengths of object-centric representations for sample-efficient behavior learning, especially in tasks that require understanding object properties and their relations. By parsing the scene into individual objects, PlaySlot is able to generalize more effectively from limited and noisy demonstrations and infer complex behaviors, which are challenging for models relying on monolithic, holistic representations.
> > >
> > > Thanks again for your thoughtful and constructive comments and engagement throughout the review process. We sincerely hope that our response has helped clarify the remaining concerns, and we would be very grateful for your consideration in increasing your score.

---

### Official Review · Reviewer_19cQ · 2025-03-09

**Overall Recommendation:** 2

**Summary:**

This paper introduces PlaySlot, an object-centric video prediction model that learns inverse latent dynamics for controllable future frame forecasting and can be used in downstream tasks. PlaySlot infers object representations and latent actions from unlabeled video sequences instead of action annotations, leveraging these representations to predict possible futures. To achieve that, the model integrates an inverse dynamics module to capture scene dynamics and a conditional object-centric predictor for forecasting. Through experiments on various environments, the authors demonstrate that PlaySlot outperforms both stochastic and object-centric baselines in video prediction accuracy, while enabling robot behavior learning from demonstrations.

**Claims And Evidence:**

Although the authors claim that the inferred latent actions enable sample-efficient learning of robot behaviors from unlabeled video demonstrations, the action decoder still requires ground-truth actions for training, which contradicts the claim of a fully unsupervised approach.

**Essential References Not Discussed:**

As far as I am concerned, the paper discusses the most relevant prior work.

**Experimental Designs Or Analyses:**

In the comparison presented in Table 1, PlaySlot infers latent actions using future frames before utilizing them for predicting subsequent frames, which raises concerns about potential information leakage. This approach may grant an advantage over baseline methods that do not have access to future frames, calling into question the fairness of the comparison.

**Methods And Evaluation Criteria:**

Yes

**Other Comments Or Suggestions:**

N/A

**Other Strengths And Weaknesses:**

### Strength
- The authors provide extensive experiments and model analysis.
- The authors explore action-conditioned object-centric video prediction.

### Weakness
- The main weaknesses are listed in the previous sections.
- While the paper claims novelty in combining object-centric video prediction with latent action models, both object-centric prediction and latent action learning in conditional video prediction have been previously explored.
- Expert data is required for behavior learning.
- The GridShapes environment seems visually too simple.

**Questions For Authors:**

1. Equation 7: Can you explain why the latent action is the difference between two consecutive embeddings?
2. How is the meaning of latent actions mapped to action numbers, as shown in Figures 5 and 6?
3. Since the model uses a pretrained SAVi, does each frame’s representation solely encode visual information from that frame? If so, would two consecutive frames be insufficient to infer an action?

**Relation To Broader Scientific Literature:**

This paper builds on the work in object-centric learning, particularly methods that decompose visual scenes into structured representations using slot-based models such as SaVi. Additionally, it relates to object-centric video prediction, which has been explored in works such as SlotFormer and OCVP, but differs by emphasizing controllability through inferred actions.  PlaySlot extends these approaches by incorporating inverse latent dynamics, drawing inspiration from prior research in learning latent actions from unlabeled videos.

**Theoretical Claims:**

N/A

---

> ### Author Rebuttal · Authors · 2025-04-01
>
> Thanks for  the  constructive comments. We are delighted that you consider our experimental section and model analysis as a strength.
> Below we address the highlighted weaknesses and questions, and clarify some misunderstandings.
>
> **The action decoder still requires ground-truth actions for training, which contradicts the claim of unsupervised learning of robot behaviors from unlabeled demonstrations.**
>
> We respectfully disagree with this statement and would like to provide further clarification.
>
> The action decoder is only needed to decode and execute latent actions in the simulator, both for qualitative evaluations (e.g. Fig. 7 ‘Sim. Actions’) and quantitative evaluation (i.e. measuring success rate as in Appendix E.2.).
> Nevertheless, PlaySlot learns robot behaviors in a completely unsupervised manner within its latent imagination. The policy model is trained in isolation to predict latent actions inferred by the inverse dynamics model from unsupervised expert demonstrations, without relying on ground-truth actions entirely.
>
> In summary, PlaySlot learns robot behaviors and to infer inverse dynamics from unlabelled demonstrations in an unsupervised manner, and the ground-truth labels are solely needed to train an action decoder for evaluation purposes only.
>
>
> **Regarding potential information leakage when using future frames.**
>
> We follow the same evaluation protocol as CADDY, where the model is first used to infer latent actions using future frames, and then it uses those same latent vectors to forecast/reconstruct the future video frames given the inferred latent actions and a small number of seed frames.
> This protocol is consistently applied across all controllable and stochastic models (CADDY, SVG, and PlaySlot), ensuring a fair comparison.
>
> Furthermore, we mitigate information leakage by enforcing an information bottleneck on the latent actions, ensuring they capture only scene dynamics rather than the full target scene state. This prevents the model from directly encoding future frame details into the latent actions.
>
>
> **Equation 7: Can you explain why the latent action is the difference between two consecutive embeddings?**
>
> The latent action is modeled as the difference between two consecutive scene dynamics’  embeddings, thus capturing the state transformation between frames.
>
> Since each embedding independently encodes information from its respective frame, their difference isolates the temporal change, which corresponds to the performed action.
>
> Additionally, we assume that action embeddings are Gaussian-distributed and independent, ensuring that their difference also follows a Gaussian distribution, facilitating a probabilistic formulation for action prediction.
>
>
>
> **How is the meaning of latent actions mapped to numbers as shown in Figs 5 & 6?**
>
> The numbers correspond to the action prototype that represents such movement. We simply add a text-label describing the action (e.g. ‘move up’) to make visualizations clearer.
>
>
>
> **Since the model uses a pretrained SAVi, does each frame’s representation solely encode visual information from that frame? If so, would two consecutive frames be insufficient to infer an action?**
>
> Yes, each frame’s representation encodes information from that frame alone. Therefore, we need our Inverse Dynamics module to infer a latent action that encodes the scene dynamics between each pair of consecutive frames.
> We indeed only use two consecutive frames to infer the latent action executed between them.
>
>
> **Weakness: GridShapes seems visually too simple.**
>
> GridShapes is indeed a visually simple dataset. However, to correctly solve this task, a method must be able to jointly model the motion of multiple moving objects.
> We find out (Fig. 4) that PlaySlot successfully models the motion of multiple (e.g. 5) objects, whereas baselines such as CADDY fail for more than two objects despite the visual simplicity of the dataset.
> This highlights the potential of object-centric world models for dynamic scenarios with multiple moving agents.
>
> In the following link, we show qualitative rollouts on GridShapes with 2 & 3 objects. We can see how the CADDY baseline fails to jointly model the motion of 3 shapes, whereas PlaySlot solves the task.
>  - https://anonymous.4open.science/r/Rollouts-7F68/
>
>
> **Weakness: Expert data is required for behavior learning.**
>
> Similar to latent action models, e.g. Genie (Bruce et al. ICML 2024) or LAPO (Schmidt & Jiang. ICLR 2024), PlaySlot indeed requires expert demonstrations to learn robot behaviors.
> However, as shown in Appendix E.2, we demonstrated that PlaySlot can learn behaviors from unlabelled video demonstrations (without access to simulators and without action information) in a sample efficient manner.
> To make this clearer, we will include the experiments showing sample-efficient behavior learning into the main text for the final version.

---

> > ### Comment · Reviewer_19cQ · 2025-04-09
> >
> > Thank you for your rebuttal. I still have concerns regarding the fairness of the comparison. The paper mentions using the original implementation of SVG in the appendix. As I understand it, SVG is a typical video prediction method in which the model does not have access to future frames during prediction. I am still not convinced about the protocol used in the paper, as it seems more like a reconstruction task rather than true prediction, which raises doubts about the validity of the comparison.
> >
> > Regarding the need for ground-truth actions, I understand the authors’ point. However, I believe the paper introduces ambiguity and potentially overclaims by suggesting that ground-truth actions are unnecessary, as also noted by reviewer nim7. The paper claims that ground-truth actions are not required, but when it comes to deploying the learned robot behavior, they are still necessary, which may mislead readers.

---

> > > ### Author Response · Authors · 2025-04-09
> > >
> > > Thank you for engaging with our rebuttal and for taking the time to consider our clarifications. We truly appreciate your comments and your openness to discussion.
> > >
> > > We would like to respectfully emphasize that the main two concerns you raise appear to stem from slight misunderstandings of our methodology. We address each of these below in detail, and also provide further clarifications and new results that support our claims:
> > >
> > > **Fairness of comparisons and use of future frames:**
> > >
> > > We respectfully disagree with the concern regarding unfair comparisons due to the use of future frames for inferring latent actions.
> > >
> > > In our work, **we follow the exact same evaluation protocol as CADDY**, where future frames are used to infer latent actions, which are then used along with the seed frames to predict future frames.
> > > This protocol is consistently applied across all methods, including PlaySlot and the baselines, thereby ensuring a fair comparison.
> > >
> > > We agree that this evaluation setup reflects a *video reconstruction* task rather than pure *open-loop prediction*. However, this is a deliberate choice following prior work where the focus is on evaluating how well the model captures the stochastic scene dynamics and uses them to forecast future frames. This is a well-established practice and not specific to our method.
> > >
> > > **Remarks regarding SVG:**
> > >
> > > As you correctly point out, SVG is a stochastic video prediction model.
> > > Importantly, SVG includes a *posterior module* that **explicitly uses future frames as input** during training to infer latent vectors, which capture stochastic scene dynamics and guide future predictions.
> > >
> > > This same mechanism is often used at inference time as well — where the posterior is used to generate rollouts that better reflect the true underlying dynamics of the target scene. For example, SVG provides qualitative evaluations using this approach (https://sites.google.com/view/svglp/), where Approx. Posterior rollouts are shown using future-frame-informed latents.
> > >
> > > Our evaluation protocol mirrors this exact same structure: we use future frames to infer latent actions (analogous to SVG’s stochastic latents), which are then used to condition the prediction of future frames from seed frames.
> > > In both cases, future information is used to infer a compact representation of stochastic scene dynamics—not the exact future frames themselves.
> > >
> > > This form of *posterior inference with future frames* is well-established and widely accepted in the stochastic video prediction literature. As such, our evaluation setup is entirely in line with existing methodologies and **does not give PlaySlot an unfair advantage over the baselines**.
> > >
> > >
> > > **Need for ground-truth actions:**
> > >
> > > We understand your concern regarding the use of ground-truth actions and appreciate your acknowledgment of our clarification.
> > >
> > > To reiterate: **ground-truth actions are never used to train the world model, policy, or inverse dynamics modules**. They are used solely to train a lightweight action decoder for evaluation — specifically to decode latent actions into executable robot commands within the simulator for evaluation purposes only.
> > > This is consistent with standard practice in the literature, such as Genie (Bruce et al. ICML. 2024) and LAPO (Schmidt & Jiang. ICLR. 2024), where evaluation requires an action decoder to deploy policies trained in latent space.
> > >
> > > The core learning process of PlaySlot is fully unsupervised, operating entirely on unlabelled (actionless) videos. No part of learning the world model, robot behaviors or inverse dynamics requires action labels, and the resulting policy operates in latent space. We agree this point could be made clearer and will revise the paper to avoid any ambiguity.
> > >
> > > **Additional Evaluations:**
> > >
> > > To further support our claims and respond to reviewers’ comments, we conducted additional experiments during the rebuttal period:
> > >  - Qualitative comparisons (rollouts) with baselines and results on Sketchy: https://anonymous.4open.science/r/Rollouts-7F68/README.md
> > >  - Comparison of PlaySlot vs LAPO for sample-efficient behavior learning from demonstrations: https://anonymous.4open.science/r/Rollouts-7F68/BEHAVIOR_LEARNING.md
> > >
> > >
> > > We believe our work offers a meaningful contribution to the community by being the first to integrate object-centric representations with latent action models for controllable video prediction and planning. This combination enables structured, interpretable world modeling that goes beyond traditional holistic approaches. Our results demonstrate that slot-based object-centric world models are not only feasible in this setting but also highly effective—particularly in robotic tasks that require relational reasoning and explicit modeling of object interactions.
> > >
> > > Thanks again for your constructive comments and your engagement in the review process. We sincerely hope that our response has helped clarify remaining concerns, and we would be very grateful for your consideration in raising your score.

---

### Official Review · Reviewer_Cnpu · 2025-03-11

**Overall Recommendation:** 4

**Summary:**

This work introduces a novel approach for the video prediction task using object-centric representations. It proposes an InvDyn module to learn latent action embeddings and a conditional object-centric predictor to forecast future object slots. Extensive experimental results demonstrate superior performance compared to previous methods, including evaluations on a real-robot video dataset.

**Claims And Evidence:**

Yes, the results strongly support the claims.

**Essential References Not Discussed:**

No.

**Experimental Designs Or Analyses:**

The experimental design is well-structured, but it lacks an analysis of the low performance on the ButtonPress dataset.

**Methods And Evaluation Criteria:**

The method is well-founded, and the evaluation follows standard practices.

**Other Comments Or Suggestions:**

No.

**Other Strengths And Weaknesses:**

Strengths:
1. The method is novel and interesting.
2. The experiments are well-designed and robust.
3. The paper is well-written.

Weaknesses:
1. Clarification on ButtonPress performance: In Table 1, the lower performance on the ButtonPress task requires further clarification.
Comparison with prior work: In Section 3.2, what are the key differences between the proposed InvDyn module and Menapace et al. (2021)?
2. Handling multiple input types during inference: In Figure 1, can multiple input types be combined during inference? I understand the other two types, but what would a human input look like?
3. What format does it take?
4. How is it processed, considering that the outputs from the InvDyn module or a learned action policy are probabilistic and implicit, whereas human inputs (e.g., "move up" or "move down") are explicit, discrete, and symbolic?

**Questions For Authors:**

Overall, this is a solid paper. However, I have one question that many researchers in this field often face: beyond the relatively simple block-based scenarios (even on the Sketchy dataset), how can this approach generalize to more real-world domains and complex scenarios? I believe this remains the main bottleneck for slot-based methods.

**Relation To Broader Scientific Literature:**

This work presents an improvement in video prediction using object-centric learning.

**Theoretical Claims:**

No theoretical claims are proposed in this work.

---

> ### Author Rebuttal · Authors · 2025-04-01
>
> Thanks for the positive review, acknowledging our experimental design, highlighting that our method is novel and interesting, and that our paper is well-written.
> Below we address some of your questions and add clarifications about certain weaknesses:
>
> **Clarification about the lower performance on ButtonPress.**
>
> The Metaworld Button-Press task is inherently non-object-centric, i.e., there is no need to reason about object properties and interactions to press the button. Therefore, no large gains can be obtained from using a structured object-centric representation of the scene.
> Nevertheless, we demonstrate that our PlaySlot model is general and still remains competitive in non-object-centric scenarios, ranking second among the evaluated methods on ButtonPress.
>
>
> **What are the key differences between the proposed InvDyn module and Menapace et al. (2021)?**
>
> There are two main differences:
>  - *Scene representation*: CADDY models scene dynamics using a holistic representation, while our InvDyn model leverages object slot representations, enabling finer-grained object-centric reasoning.
>
>  - *Action Parameterization*: Both methods represent actions differently, leading to distinct modeling choices.
>
> A more detailed discussion of these differences is provided in the Appendix D.2.
>
>
> **About inputs provided by a human to PlaySlot**
>
> PlaySlot parameterizes an action with a discrete action prototype, which represents the action taking place, and an action variability embedding, which allows to interpolate between different motions or modify the action prototypes (e.g. changes in speed or direction).
> To allow a user to control PlaySlot, we first qualitatively measure the effect of each learned action prototype and then assign a semantic meaning (e.g. ‘move left’) to each action prototype according to the motion it represents.
> At inference time, a user can specify actions by selecting the corresponding action prototypes, while setting the action variability embedding as zero.
>
>
> **How can this approach generalize to more real-world domains and complex scenarios? I believe this remains the main bottleneck for slot-based methods.**
> Recent works, such as DINOSAUR (Seitzer et al. ICLR 2023) or SlotDiffusion (Wu et al NeurIPS 2023), show that slot-based models can be extended to handle more realistic data by using large self-supervised pretrained encoders, or powerful diffusion-based decoders.
> As discussed in the Limitations & Future Work section (Appendix A), we plan to explore in future work these architectural designs to learn robot behaviors in more complex robotic scenarios

---

> > ### Comment · Reviewer_Cnpu · 2025-04-03
> >
> > I will keep my score.

---

> > > ### Author Response · Authors · 2025-04-08
> > >
> > > We are pleased that we were able to address the reviewer’s comments and questions, and we sincerely thank the reviewer again for the constructive review and positive score.
> > >
> > > We really hope that our work motivates the community to further explore slot-based object-centric models for world modelling and planning, which is a research direction we are genuinely excited about.
> > >
> > > We sincerely appreciate your comments and the time you took throughout the review process, and we are happy that the strengths of our work came through. Thanks again!

---

### Official Review · Reviewer_R5dN · 2025-03-12

**Overall Recommendation:** 4

**Summary:**

The papers proposes PlaySlot - a novel approach to controllable video prediction that builds on previous work on object-centric learning and latent action learning. Unlike previous approaches to video prediction based on object-centric learning, PlaySlot incorporates the InvDyn module for inferring latent actions between observations, which allows PlaySlot to learn in a self-supervised manner, enabling controllable generation (or planning) of multiple possible futures conditioned on different latent actions. In contrast to previous latent action learning approaches (such as CADDY or LAPO), PlaySlot infers object representations as slots and models the dynamics in terms of them via the transformer-based predictor. The authors use SAVi to infer slots and adapt the architecture of forward dynamics and inverse dynamics models to work specifically with slots. Latent actions are factorized into deterministic action prototypes and action variability continuous embeddings. Using simple robotic manipulation tasks (ButtonPress, BlockPush) and a synthetic dataset (GridShapes), the authors empirically demonstrate that PlaySlot outperforms considered baselines in terms of prediction quality. Experiments also show that the resulting latent actions are meaningful and consistent, allowing for sample-efficient fine-tuning with a small number of ground truth actions.

# update after the rebuttal

I was initially reluctant to raise the score much, but during the rebuttal the authors addressed most of my concerns either by providing experimental evidence or by describing planned changes to the text. Although the changes are large and change the claims, I consider them positive. I also saw the seriousness of the authors during the rebuttal, which gives me confidence that all changes will be included in the final version of the paper.

**Claims And Evidence:**

There are two main claims in the paper (direct quotes from the text):

> 1. PlaySlot outperforms several video prediction models across diverse robotic environments, while showing superior interpretability and control capabilities.
> 2. The object representations and latent actions inferred by PlaySlot can be used to learn robot behaviors from unlabeled video demonstrations sample efficiently.

Firstly, it seems to me that the experiments presented do not allow the authors to claim superiority "across different robotic environments", as only two simple robotic environments were used, and GridShapes is not in the robotic domain. Furthermore, on ButtonPress PlaySlot does not outperform the baseline SVG (and this is just one of many tasks in MetaWorld, and not the most difficult), and even on BlockPush, which is specifically designed to favour object-centric representations, PlaySlot does not outperform the baseline by much. However, it is difficult to judge here as no standard deviation or confidence intervals were provided. Similarly, for the most interesting benchmark - real-world robot videos (Sketchy) - only PlaySlot results are provided, without the baselines and metrics.

Secondly, there is currently no evidence in the main text that PlaySlot can be used to learn robot behavior in a sample efficient manner, as only a qualitative evaluation is provided. However, there are such experiments in the Appendix, and they are quite convincing, at least for the environments tested. However, they don't include LAPO as a baseline, which I find odd given its large influence on latent action learning research and its overall simplicity. What if, despite its simplicity, it learns latent actions more suitable for efficient fine-tuning even on BlockPush?

Thus, I think that the current main claims are not sufficiently supported by the evidence provided. The authors should either provide more evidence (more than one environment for each type, clearly state number of random seeds used, provide metrics of variation, baseline metrics on Sketchy, bring experiments showing sample-efficient fine-tuning into the main text and compare with the LAPO), or remove the exaggerated claims to match the experiments provided. I understand that there may be limitations in computational resources, but this does not justify overclaiming. I still find the proposed approach novel, exciting and potentially useful. I may increase the score if some of the comments are addressed.

Some other minor claims:

> We further show how PlaySlot effectively captures precise robot actions and seamlessly scales to scenes with multiple moving objects or to real-world robotics data, outperforming several stochastic and controllable video prediction baselines
>

As noted above, there are no comparisons with baselines on real world robotics data.

> However, both CADDY and Genie operate on holistic scene representations, which are limited for tasks that require relational reasoning, often struggle to model object relationships and interactions, and require human supervision to generalize to scenes with multiple moving agents.
>

Currently there are no references in the text which support this claim.

> This approach shares similarities with Schmidt & Jiang (2024). However, whereas their method learns policies for simple games with a small discrete set of actions …
>

If I'm interpreting the numbers correctly in the Appendix A.4 from the LAPO paper, they used latent actions of dimension 128, which were then reshaped to 8 discrete ones, each with a codebook size of 64. Which gives a total of 64^8 actions. I wouldn't call that small.

**Essential References Not Discussed:**

To my knowledge, paper cites all the relevant papers essential for understanding the context and key contributions.

**Experimental Designs Or Analyses:**

Apart from the concerns discussed in 'Methods and evaluation criteria', I think the only concern with experimental design is too much focus on qualitative comparisons.

First, it's pretty hard to tell if it's not cherry picking, which is quite possible even without explicit intent. Second, it's quite repetitive, better to demonstrate it once and use the remaining space for more valuable experiments (e.g. sample-efficient fine-tuning with ground truth actions).

Finally, while visual quality is important for video generation, it is less important for world modelling and planning (unless the goal is to train policies in image space rather than compact latent space, but this is rare). A world model can be globally inaccurate and still be better for planning and downstream performance than a more accurate one (see [1]).

Therefore, to support the claim that PlaySlot captures precise robot actions useful for planning and fine-tuning, which is what most practitioners would value, it's necessary to include fine-tuning experiments in the main part of the paper.

As for the ablations, I don't have any concerns. I think they show the right things and actually improve the understanding of the proposed method.

References:

1. Lambert, N., Amos, B., Yadan, O., & Calandra, R. (2020). Objective mismatch in model-based reinforcement learning. *arXiv preprint arXiv:2002.04523*.

**Methods And Evaluation Criteria:**

Despite the fact that the evaluation is limited to a few simple domains, they are fully suitable for potentially verifying the claims of the article. The benchmarks considered include domains that require non-object and object-centred reasoning, as well as real-world robotics data. The experiments also use standard and widely used metrics for video prediction.

However, the evaluation itself is questionable. Nowhere in the paper (including the appendix) is it stated how many random seeds were used for the experiments. Results are also reported without standard deviation or confidence intervals (for video prediction experiments as well as for performance after fine-tuning with ground truth actions).

The hyperparameter search process is not explicitly documented. Was there one, or did all methods use the default hyperparameters? If so, was it explicitly considered that all methods should use the same enumeration budget? Were the methods explicitly balanced in the number of weights and gradient updates? I think the authors should state this explicitly and include a table with the number of trainable weights for each method.

**Other Comments Or Suggestions:**

I do not have any other comments or suggestions.

**Other Strengths And Weaknesses:**

Strengths:

- The paper is clearly written and presents the method in a way that's easy to understand.
- The method is novel in the sense that, to my knowledge, there is no other work that combines object-centred representations with latent action learning.
- The related work section provides all the necessary context and does not overlook important papers in both fields.

Weaknesses:

- As discussed in the sections above, the empirical evidence is weak.
- The evaluation protocol is not clearly explained. Information on number of random seeds, standard deviations or confidence intervals, hyperparameter search is not provided.

**Questions For Authors:**

1. Does ablation on action representations include variants without sampling/stochasticity (e.g. continuous part)? Is there really any benefit to sampling? LAPO claims that it can capture stochasticity, and they do not sample latent actions.
2. Where these experiments reported?

    > We empirically verify that the information bottleneck enforced by vector quantization achieves comparable performance to the one proposed by (Menapace et al., 2021), while requiring significantly fewer hyper-parameters.
    >

**Relation To Broader Scientific Literature:**

PlaySlot is a novel combination of object-centric learning, i.e. SAVi, and latent action learning, pioneered by the seminal work on Imitating Latent Policies from Observation (ILPO) [1], and recently significantly improved by LAPO [2] and LAPA [3]. PlaySlot, on the other hand, borrows many design decisions from CADDY [4]. However, the empirical evidence provided is rather weak and does not allow to conclude that the PlaySlot approach is definitely a better way for latent action learning, e.g. compared to LAPA, which also uses the robotics domain. Without such evidence, the main contribution is rather incremental.

References:

1. Edwards, A., Sahni, H., Schroecker, Y., & Isbell, C. (2019, May). Imitating latent policies from observation. In *International conference on machine learning* (pp. 1755-1763). PMLR.
2. Schmidt, D., & Jiang, M. (2023). Learning to act without actions. arXiv preprint arXiv:2312.10812.
3. Ye, S., Jang, J., Jeon, B., Joo, S., Yang, J., Peng, B., ... & Seo, M. (2024). Latent action pretraining from videos. *arXiv preprint arXiv:2410.11758*.
4. Menapace, W., Lathuiliere, S., Tulyakov, S., Siarohin, A., & Ricci, E. (2021). Playable video generation. In *Proceedings of the IEEE/CVF Conference on Computer Vision and Pattern Recognition* (pp. 10061-10070).

**Theoretical Claims:**

The paper does not present any new theoretical results or claims. After careful examination, I do not see any flaws or errors in the formulas used to present the approaches, e.g. SlotAttention and other components of the PlaySlot pipeline.

---

> ### Author Rebuttal · Authors · 2025-04-01
>
> Thanks for the constructive review and for finding our proposed approach novel, exciting and potentially useful.
>
> Below we address your questions and main comments. We will address other comments (e.g. remove exaggerated claims and add missing references) in the final version of the paper. Furthermore, we will reduce the number of qualitative evaluations on the main text, and include the sample-efficient Behavior Learning experiment instead.
>
>
> **Lack of quant. evaluation and baselines on Sketchy**
>
> We quantitatively evaluate PlaySlot, as well as SVG and CADDY on the real-robot Sketchy dataset. The results are:
>
> | | **PSNR** | **SSIM** | **LPIPS** |
> | --- | --- | --- | --- |
> | **SVG** | *24.24*| *0.813* | 0.076 |
> | **CADDY** | 23.65 | 0.809 | **0.059** |
> | **PlaySlot** | **24.43** | **0.815** | *0.063* |
>
> We observe that all methods achieve an overall similar performance on this dataset, with PlaySlot slightly outperforming the baselines. We show qualitative comparisons and prediction rollouts for all methods in the following link: https://anonymous.4open.science/r/Rollouts-7F68/
>
> Despite not outperforming the baselines by a significant margin, these results show that object-centric models like PlaySlot can be applicable for playable video prediction and as a world model on real robot data. In future work, we plan to extend our method with more expressive encoders and decoders (see Appendix A) to apply PlaySlot on more complex real robotic scenarios.
>
>
>
> **Lack of LAPO baseline for behavior learning**
>
> We are currently working on training and training LAPO on our BlockPush environment. We hope to provide a comparison between PlaySlot and LAPO by the end of the rebuttal period.
>
>
> **Some additional differences with LAPO/LAPA**
>
> LAPO & LAPA are primarily designed and optimized for latent-action pretraining in the context of learning policies for 2D games (LAPO) or robot behaviors (LAPA). However, these models are not designed for autoregressive future prediction, limiting their effectiveness as world models.
>
> In contrast, PlaySlot incorporates a powerful object-centric world model, which enables the generation of future trajectories conditioned on latent actions.
>
>
>
> **Does ablation on action representations include variants without stochasticity (e.g. continuous part)? Is there really any benefit to sampling?**
>
> Our ablation includes a comparison with an oracle model that does not perform any sampling. However, the remaining PlaySlot variants (hybrid, continuous, discrete) sample latent actions $\hat{\mathbf{z}_t}$ from a learned Gaussian.
>
> Inspired by literature in stochastic video prediction, e.g. SVG (Denton & Fergus 2018), CADDY (Meanapace et al. 2021) or SLRVP (Franceschi et al. 2020), we use a stochastic formulation to learn a structured latent space that captures uncertainty and enables controllable video prediction.
>
>
> **Were these experiments reported? ',.. information bottleneck enforced by vector quantization achieves comparable performance to the one proposed by CADDY while...'**
>
> No, these experiments were not included in the paper. We conducted early experiments where both variants performed comparably and decided to stick with our approach.
> The method proposed by Menapace et al. involves numerous hyper-params (e.g., many loss weights and Gumbel-Softmax temperature scheduling), which impact performance. A poor choice of these hyper-params often leads to suboptimal training runs where multiple latent actions represent the same motion, or where multiple action prototypes did not encode any meaningful representations.
> We found that tuning the hyper-params for all datasets would be challenging and expensive. Therefore, we favored our approach, which is more practical.
>
>
> **Training and Compute Hyper-parameters**
>
> To ensure a fair comparison, we balance the number of learnable params and compute requirements for all methods. Below, we report the number of trainable params for each model:
>  SVG:  18.84M; CADDY:  9.34M; SlotFormer:  3.97M (2.96M for predictor & 1.01M for SAVi);  OCVP: 4.86M  (3.85M for predictor & 1.01M for SAVi); PlaySlot: 8.49M  (3.19M for InvDyn, 4.27M for cOCVP & 1.01M for SAVi).
>
> Regarding hyper-param tuning, we use the following structured approach:
>  - For CADDY and SVG we start with the hyper-params from the BAIR dataset and manually refine them for our datasets.
>  - For SlotFormer, OCVP and PlaySlot, we use the default OCVP hyper-params as a baseline and further adjust them for optimal performance on our datasets.
>
> While we acknowledge that a more exhaustive hyper-param search could yield better results, conducting extensive searches for all methods was infeasible due to compute budget and hardware constraints.
>
> Regarding the number of random seeds. We run the behavior learning experiments (Appendix E.2) with 3 different seeds and report now the mean success rate and standard deviation. The results on ButtonPress are provided in: https://anonymous.4open.science/r/Rollouts-7F68/

---

> > ### Comment · Reviewer_R5dN · 2025-04-03
> >
> > I thank the authors for taking into account the feedback and putting time and effort into improving the paper. I see this as a positive development and believe that the changes will strengthen the claims.
> >
> > I raise my score to 3. As these are quite large changes that I won't be able to validate during this review, I can't raise my score any higher.

---

> > > ### Author Response · Authors · 2025-04-08
> > >
> > > We are pleased that we were able to address the reviewer’s comments and questions, and we sincerely thank the reviewer for recognizing our efforts in the rebuttal and for increasing the score.
> > >
> > > We believe that our work contributes meaningfully towards understanding the utility of object-centric world models for controllable video prediction and planning. While our proposed method may not universally outperform all baseline approaches, we strongly believe that slot-based object-centric representations offer a valuable complement to world models and latent action models — particularly in robotic tasks that demand relational reasoning or explicit modeling of object interactions.
> > >
> > > To further emphasize the strengths of PlaySlot and the use of a slot-based structured latent space, as well as motivated by your review, we have compared the performance of our proposed PlaySlot model with LAPO (Schmidt & Jiang. ICLR. 2024) on both the *ButtonPress* and *BlockPush* environments.
> > > Specifically, we evaluated and compared their ability to sample-efficiently learn robot behaviors from a limited number of expert demonstrations. A detailed discussion of these experiments and results is provided in:
> > >  - https://anonymous.4open.science/r/Rollouts-7F68/BEHAVIOR_LEARNING.md
> > >
> > > Below, we summarize the key findings:
> > >
> > >  - PlaySlot and LAPO perform comparably on the *ButtonPress* environment, which is less reliant on object-centric reasoning.  Nonetheless, PlaySlot achieves slightly better sample-efficiency and higher performance than LAPO across most data regimes.
> > >  - On the more challenging *BlockPush* task — where understanding object properties and their relations is crucial — PlaySlot consistently outperforms LAPO by a large margin across all data regimes, demonstrating much stronger sample-efficiency and substantially higher performance.
> > >
> > > These results highlight the strengths of object-centric representations for sample-efficient behavior learning, especially in tasks that require understanding object properties and their relations. By parsing the scene into individual objects, PlaySlot is able to generalize more effectively from limited and noisy demonstrations and infer complex behaviors, which are challenging for models relying on monolithic, holistic representations.
> > >
> > > We would like to thank you again for your thoughtful and constructive comments and engagement throughout the review process. We sincerely hope that our response has addressed any remaining concerns, and we would greatly appreciate your consideration in raising your score.

---

### Decision · Program_Chairs · 2025-05-01

**Decision:**

Accept (poster)

**Comment:**

This paper received overall positive reviews, with reviewers finding the proposed PlaySlot framework to be a novel and meaningful contribution, combining object-centric video prediction with latent action learning in a way that enables controllable future forecasting from actionless videos. Several strengths were noted, including the clear writing and thoughtful integration of inverse dynamics with slot-based representations. Concerns were raised about limited evaluations (e.g., only simple robotic environments, missing baselines on real-world data), but the authors addressed these in the rebuttal and committed to revising the claims to better reflect the scope of the experiments.

The AC agrees with reviewers that PlaySlot represents a valuable step forward in controllable video prediction. The authors are encouraged to clarify evaluation details, moderate the paper's claims, and incorporate the discussion from the discussion phase into the camera-ready version.